# Synthesis, Antiproliferative Evaluation and QSAR Analysis of Novel Halogen- and Amidino-Substituted Benzothiazoles and Benzimidazoles

**DOI:** 10.3390/ijms232415843

**Published:** 2022-12-13

**Authors:** Valentina Rep Kaulić, Livio Racané, Marijana Leventić, Domagoj Šubarić, Vesna Rastija, Ljubica Glavaš-Obrovac, Silvana Raić-Malić

**Affiliations:** 1Department of Organic Chemistry, Faculty of Chemical Engineering and Technology, University of Zagreb, Marulićev trg 20, 10000 Zagreb, Croatia; 2Department of Applied Chemistry, Faculty of Textile Technology, University of Zagreb, Prilaz Baruna Filipovića 28, 10000 Zagreb, Croatia; 3Department of Medicinal Chemistry, Biochemistry and Laboratory Medicine, Faculty of Medicine Osijek, University Josip Juraj Strossmayer of Osijek, Josipa Huttlera 4, 31000 Osijek, Croatia; 4Faculty of Agrobiotechnical Sciences Osijek, Josip Juraj Strossmayer University of Osijek, Vladimira Preloga 1, 31000 Osijek, Croatia

**Keywords:** benzothiazoles, benzimidazoles, anticancer, QSAR, cell cycle perturbation, mitochondrial membrane potential

## Abstract

Syntheses of 6-halogen-substituted benzothiazoles were performed by condensation of 4-hydroxybenzaldehydes and 2-aminotiophenoles and subsequent *O*-alkylation with appropriate halides, whereas 6-amidino-substituted benzothiazoles were synthesized by condensation of 5-amidino-2-aminothiophenoles and corresponding benzaldehydes. While most of the compounds from non-substituted and halogen-substituted benzothiazole series showed marginal antiproliferative activity on tested tumor cell lines, amidino benzazoles exhibited stronger inhibitory activity. Generally, imidazolyl benzothiazoles showed pronounced and nonselective activity, with the exception of **36c** which had a strong inhibitory effect on HuT78 cells (IC_50_ = 1.6 µM) without adverse cytotoxicity on normal BJ cells (IC_50_ >100 µM). Compared to benzothiazoles, benzimidazole structural analogs **45a**–**45c** and **46c** containing the 1,2,3-triazole ring exhibited pronounced and selective antiproliferative activity against HuT78 cells with IC_50_ < 10 µM. Moreover, compounds **45c** and **46c** containing the methoxy group at the phenoxy unit were not toxic to normal BJ cells. Of all the tested compounds, benzimidazole **45a** with the unsubstituted phenoxy central core showed the most pronounced cell growth inhibition on THP1 cells in the nanomolar range (IC_50_ = 0.8 µM; SI = 70). QSAR models of antiproliferative activity for benzazoles on T-cell lymphoma (HuT78) and non-tumor MDCK-1 cells elucidated the effects of the substituents at position 6 of benzazoles, demonstrating their dependence on the topological and spatial distribution of atomic mass, polarizability, and van der Waals volumes. A notable cell cycle perturbation with higher accumulation of cells in the G_2_/M phase, and a significant cell increase in subG0/G1 phase were found in HuT78 cells treated with **36c**, **42c**, **45a**–**45c** and **46c**. Apoptotic morphological changes, an externalization of phosphatidylserine, and changes in the mitochondrial membrane potential of treated cells were observed as well.

## 1. Introduction

Due to its high prevalence, complexity, and dangerous mortality rate, cancer has been listed as the second leading cause of death worldwide, accounting for nearly 10 million deaths in 2020, or nearly one in six deaths [1]. Since 2020, cancer chemotherapy has been severely impacted by the COVID-19 pandemic, resulting in delays in diagnosis and treatment, which may lead to an increase in advanced stage disease and ultimately increased mortality [2]. Given the limited efficacy of currently available anticancer drugs and the rapid development of resistance due to genetic mutations of accessible targets, there is a growing need to design and develop new drug candidates with higher efficacy and lower toxicity [3].

As in recent years, many drugs authorized in 2020 contain nitrogen aromatic heterocycles [4]. Nitrogen aromatic heterocycle-based compounds exhibit anticancer effects through either cell growth arrest or induction of cell differentiation and apoptosis [5]. Owing to structure similarity to the natural occurring purines, benzimidazole and benzothiazole derivatives are useful scaffolds in drug discovery of anticancer agents [6,7,8,9,10]. Some benzimidazole-based compounds such as abemaciclib, bendamustine, crenolanib, dovitinib, galeterone, glasdegib, liarozole, nocodazole, pracinostat, selumetinib, and veliparib have been approved for the treatment of various cancers [11]. Moreover, benzimidazole [11,12,13] and benzothiazole [14,15,16] hybrids exhibit dual or multiple antiproliferative activities and, therefore, have the potential to increase efficacy and overcome cancer drug resistance. The 2-arylbenzothiazole derivative CJM-126 (I) (Figure 1) was found to exhibit potent growth inhibition on human-derived breast carcinoma MCF-7 cell lines, including estrogen receptor-positive MCF-7^wt^ cells with an IC_50_ value < 0.001 µM [17,18]. Another example of highly potent anticancer agent is benzothiazole analog MKT-077 (II), a water-soluble rhodocyanine dye, acting as an inhibitor of heat shock protein 70 (Hsp 70) [19]. 6-Fluorobenzothiazole (PMX 610) (III) exhibited potent and selective in vitro antitumor properties in human cancer cell lines (e.g., colon, non-small cell lung, and breast subpanels) [20,21,22], while compound 5F203 (IV**)** proved efficient in nanomolar range against MCF-7 cells. Its prodrug Phortress (V), with high bioavailability tested in clinical trials, showed activity against renal, breast, ovarian and colorectal solid carcinoma. Its mechanism of action involves in vivo hydrolysis to release 5F203 (IV), which is further metabolized by the P450 enzyme CYP1A1 to a highly reactive species, which attacks and breaks DNA strands, ultimately leading to cell death [9,17,23,24].

We have found that, among diverse benzimidazole amidines, imidazolyl benzimidazole with benzyl-1,2,3-triazole VI (Figure 2) exhibits potent growth-inhibitory activity against non-small cell lung cancer A549 cells, which was associated with induction of apoptosis and primary necrosis [25]. 1-(*p*-Chlorophenyl)-1,2,3-triazole-tagged benzimidazole VII also showed selective inhibitory effect on A549 cells, inducing p38 MAPK- and NF-κB-mediated apoptosis [26]. Moreover, in the series of benzothiazole amidines, VIII and IX, which exhibited strong antiproliferative effect on colorectal cancer SW620 and MCF-7 cell lines, respectively, also showed noncovalent interaction with 6-amidinobenzothiazole ligands, demonstrating both minor groove binding, and intercalating mode of DNA interaction [27].

In view of the biological importance of benzothiazole and benzimidazole pharmacophores, and as a part of ongoing research focused on the development of new anticancer agents [25,26,27], we have designed and synthesized a small library of 2-aryl-substituted benzothiazole and benzimidazole entities aiming to evaluate their antiproliferative activities (Figure 2). In this context, halogen and amidino benzothiazoles were linked via phenoxymethylene spacer to diverse aromatic subunits, to ensure the distribution of highly hydrophilic and hydrophobic parts of the structure. Antiproliferative effects of novel 6-halogen, 6-imidazolyl and 6-pyrimidinyl benzothiazole derivatives, and previously prepared benzimidazoles [28], were evaluated on selected human tumor cell lines. The results of quantitative structure–activity relationship (QSAR) analysis on T-cell lymphoma (HuT78) and Madin–Darby canine kidney cells (MDCK1) were compared and discussed. Imidazolyl benzothiazole **36c**, and pyrimidinyl benzimidazoles **42c**, **45a**, **45b**, **45c**, and **46c** with potent and selective antiproliferative activity were also evaluated in cell-cycle perturbation and mitochondrial membrane potential assays.

## 2. Results and Discussion

### 2.1. Chemistry

A library of 22 previously published pyrimidinyl benzimidazole [28] derivatives was expanded with novel 55 halogen- and amidine-substituted benzothiazole analogs, prepared by a multi-step synthetic route as shown in Figure 1, Figure 2, Figure 3 and Figure 4. Benzoyl (**15a**–**15c, 16a**–**16c, 17a**–**17c**) and picolyl (**18a**–**18c, 19a**–**19c, 20a**–**20c**) 6-halogen-substituted and 6-unsubstituted benzothiazoles were prepared by a four-step synthesis. The key intermediates **9a**–**9c, 10a**–**10c** and **11a**–**11c** were prepared in moderate reaction yields (33%–78%) by condensation of corresponding 4-hydroxybenzaldehydes **8a**–**8c** and 2-aminotiophenoles **5**–**7** using sodium metabisulfite (Na_2_S_2_O_5_) as a mild oxidant (Figure 1). A base-promoted *O*-alkylation of 2-(4-hydroxyphenyl)benzothiazoles (**9a**–**9c, 10a**–**10c**) with corresponding halides gave target 6-halogen-substituted benzothiazole derivatives with benzoyl (**15a**–**15c, 16a**–**16c**) and picolyl (**18a**–**18c, 19a**–**19c**) units, and introduced the phenoxymethylene linker in low to moderate yields (22–66%), whereas 6-unsubstituted benzothiazole analogs **17a**–**17c** and **20a**–**20c** were isolated in good to high yields (41–81%). *O*-Propargylated intermediates (**12a**–**12c, 13a**–**13c, 14a**–**14c**) for the synthesis of 1,2,3-triazolyl linked 2-arylbenzothiazole derivatives (**21a**–**26a**, **21b**–**26b** and **21c**–**26c**) were prepared. To evaluate the effect of triazole moieties on biological activity, 1*H*-1,2,3-triazole benzothiazole derivatives **21a**–**21c**, **22a**–**22c** and **23a**–**23c** were synthesized via a regioselective copper(I) catalyzed cycloaddition, with copper(I) iodide and trimethylsilylazide, while benzothiazoles **24a**–**24c**, **25a**–**25c**, and **26a**–**26c** with 1-benzyl-1,2,3-triazole unit were obtained by the one-pot click reaction with benzyl azide formed in situ using Cu(II) acetate as a catalyst (Figure 2).

To further explore the influence of the amidino substituent in C-6 of the benzothiazole core and to improve solubility, in addition to a series of pyrimidinyl benzimidazole derivatives prepared according to the procedure previously reported by our group [28], a series of 6-imidazolyl **38a**–**38c**, **39a**, **39c**, **40a**–**40c**, **41a**–**41c** and 6-pirimidinyl **42a**, **42b**, **43c**, **44a**–**44c**, **45a**, **45c** benzothiazoles was synthesized as shown in Figure 3 and Figure 4.

Firstly, benzaldehyde precursors **27a**–**27c**, **28a**–**28c**, **and 29a**–**29c** were obtained through *O*-alkylation of 4-hydroxybenzaldehydes. Followed by a copper(I)-catalyzed Huisgen 1,3-dipolar cycloaddition reaction of benzaldehydes **27a**–**27c**, 1,2,3-triazole-substituted benzaldehydes **30a**–**30c**, **31a**–**31c** were obtained [28]. Amidino-substituted 2-aminothiophenole **32** and **33** were prepared from 6-cyanobenzothiazole by the Pinner method [29,30]. Benzothiazoles **34a**–**34c**, **35a**, **35c**, **36a**–**36c** and **37a**–**37c** were prepared by cyclocondensation of amidino-2-aminothiophenole **32** with benzaldehyde precursors (**27a**–**27c**, **28a**–**28c**, **29a**–**29c**) in acetic acid, while the benzothiazoles **38a**, **38b**, **39c**, **40a**–**40c**, **41a** and **41c** [31] were synthesized from amidino-2-aminothiophenole **33** with corresponding benzaldehydes. Finally, targeted amidino-substituted benzothiazole hydrochlorides were prepared by an acid-base reaction.

### 2.2. Evaluation of Antiproliferative Activity

The antiproliferative activities of novel benzothiazole derivatives (**15a**–**26a**, **15b**–**26b**, **15c**–**26c**, **34a**–**38a**, **40a**, **41a**, **34b**, **36b**–**38b**, **40b**, **34c**–**37c**, and **39c**–**41c** on human tumor cell lines, including cervical adenocarcinoma (HeLa), colon adenocarcinoma (CaCo-2), T-cell lymphoma (HuT78), and non-tumor Madin–Darby canine kidney (MDCK-1) cells and human fibroblasts (BJ) are presented in Table 1 and Table 2 as well as in Appendix A. 5-Fluorouracil (5-FU) was used as the reference drug. In order to assess the impact of the benzimidazole moiety as benzothiazole bioisoster on the antiproliferative activity and to compare inhibitory effects of benzothiazoles and their benzimidazole analogs, activity of previously synthesized amidino benzimidazoles **42a**–**49a**, **42b**–**45b**, **47b**–**49b**, **42c**–**46c**, **48c** and **49c** was also evaluated (Table 1, Table 2, Appendix A).

According to the results presented in the Table 1, the majority of compounds from non-substituted and halogen-substituted benzothiazole series showed marginal antiproliferative activity on tested tumor cell lines. From benzothiazoles without 1,2,3-triazole moiety **15a**–**20a**, **15b**–**20b**, and **15c**–**20c**, benzothiazoles with benzoyl moiety showed to be less active compared to their analogs with the picolyl aromatic unit. In line, 6-chlorobenzimidazole **18a** containing pyridinyl exhibited the best inhibitory activity (IC_50_ = 13.2 µM) on T-cell lymphoma (HuT78) cells, while benzothiazole **20b**, with fluorine attached at phenyl central unit, had moderate activity on CaCo-2 cells (IC_50_ = 28.7 µM), which was lower to that of 5-FU. Among 4-(1,2,3-triazolylmethoxy)phenyl benzothiazoles **21a**–**26a**, **21b**–**26b** and **21c**–**26c**, compounds **21a**–**21c**, **22a**–**22c** and **23a**–**23c** with terminal 1*H*-1,2,3-triazole ring exhibited higher activity than those with 1-benzyl-1,2,3-triazole **24a**–**24c**, **25a**–**25c** and **26a**–**26c**. Although cytotoxic in contact with normal MDCK-1 cells, these compounds exhibited only marginal inhibitory effects on growth of normal human fibroblasts (BJ). 6-Chlorobenzothiazoles **21a** and **21b** and 6-fluorobenzothiazole **22b** had the best activity on HuT78 cells (**21a**: IC_50_ = 6.8 µM, **21b**: IC_50_ = 3.6 µM, **22b**: IC_50_ = 9.1 µM). In comparison, 5-FU did not exhibit antiproliferative effect on HuT78 cells. Overall, our data suggest increased antiproliferative activity of compounds with fluorine substituent attached at aromatic central unit, while methoxy group-substituted entities demonstrated decreased inhibition efficiency in HuT78 cells. Conversely, the methoxy group in 6-unsubstituted benzothiazoles **26a**–**26c** with 1,4-disubstituted 1,2,3-triazole improved growth inhibition in both HeLa (IC_50_ = 23.3 µM) and HuT78 (IC_50_ = 23.7 µM) cell lines.

Nineteen compounds from the amidino benzothiazole and twenty two compounds from amidino benzimidazole series (Table 2) showed stronger growth inhibition than halogen- and unsubstituted benzothiazole derivatives. Imidazolyl benzothiazoles showed strong antiproliferative activity on all tested tumor cell lines. However, these compounds were also toxic on both normal cell lines, MDCK1 and BJ cells. 5-FU showed to be less cytotoxic on MDCK1 and BJ cells with values of selectivity index (SI) from 6.7 to 9.3. Interestingly, the introduction of the 1*H*-1,2,3-triazole moiety at phenyl in compounds **36a**–**36c**, resulted in less cytotoxicity against normal MDCK1 and BJ cells, while maintaining excellent growth-inhibitory effect on HuT78 cell lines (**36a**: IC_50_ = 4.4 µM; **36b**: IC_50_ = 1.8 µM; **36c**: IC_50_ = 1.6 µM) with SI of 8.8, 18.1 and 62.5, respectively, with respect to the inhibition of BJ cells (Appendix A). The methoxy group at phenyl in **36c** caused a selective and pronounced antiproliferative effect on HuT78 cells. Replacement of the 6-imidazolyl with the 6-pyrimidinyl group in **38a**, **38b**, **39c**, **40a**–**40c**, **41a** and **41c** reduced their inhibitory activity. From pyrimidinyl benzothiazole derivatives, only **38b** and **41a** expressed strong albeit not selective activity on HuT78 cells. Some 6-pyrimidinyl benzothiazoles were devoid of any antitumor activities. Thus, pyrimidinyl derivatives **40a**–**40c** and **41c** compared to their imidazolyl analogs **36a**–**36c** and **37c** did not exhibit inhibitory effect on tested cell lines, except for **40b** which showed only moderate activity against HuT78 cells (IC_50_ = 37.3 µM).

Among benzimidazole amidines **42a**–**49a, 42b**–**45b, 47b**–**49b**, **42c**–**46c**, **48c** and **49c**, the strongest antiproliferative activity on HuT78 cells was observed for analogs **45a** (IC_50_ = 4.8 µM), **45b** (IC_50_ = 5.5 µM), and **45c** (IC_50_ = 4.1 µM) with 1-benzyl-1,2,3-triazole substituent at the phenoxy core. A similar effect was observed for compound **46c** (IC_50_ = 4.1 µM), containing 1-ethylmorpholino-1,2,3-triazole side chain, with SI (compared to the inhibition of BJ cells) of 11.6, 14.9, 23.4 and 19.6, respectively (Appendix A). Their benzothiazole analogs **37a**–**37c** were more cytotoxic to human normal fibroblasts (BJ) with SI of 2.7, 2.5 and 1.6, respectively. Compared to benzothiazoles **34a** and **38a** with benzoyl moiety, benzimidazole structural analog **42a** exhibited significant and selective antiproliferative activity against T-cell lymphoma (IC_50_ = 7.0 µM on HuT78 cells, IC_50_ > 100 µM on MDCK1 and BJ cells). As shown in Table 2, benzimidazoles containing picolyl **43a**–**43c** and 1*H*-1,2,3-triazole **44a**–**44c** moiety exhibited in turn decreased inhibitory activity compared to those of benzothiazole congeners. Moreover, introduction of aliphatic moiety in benzimidazole amidines **47a**–**49a**, **47b**–**48b** and **48c**–**49c** resulted in a loss of antiproliferative activity.

Of the 77 benzazole analogs that were evaluated for their antiproliferative activity, twelve benzazoles (**36a, 36c, 38b, 39c, 42a**– **42c, 43a, 45a**–**45c** and **46c**) with marked and selective activity were chosen for evaluation on additional tumor cell lines, i.e., colorectal adenocarcinoma, metastatic (SW620), human breast adenocarcinoma (MDA-MB-231), promyelocytic leukemia (HL60) and human monocytic leukemia (THP1) cell lines (Table 3).

Interestingly, amidino benzothiazole **36c** with the methoxy group at the phenoxy core did not exhibit growth inhibition on evaluated cell lines showing selective antiproliferative activity only on HuT78 cells (Table 2 and Table 3). Similarly, amidino benzimidazole **46c** with methoxy substituent showed notable antiproliferative activity only on HuT78 cell line (Table 2), along with moderate activity on HL60 cells (Table 3). In contrast to selective antiproliferative activity of **36c** on HuT78 cells, its structural analog **36a** with the unsubstituted phenoxy unit showed inhibitory activity on additional set of tumor cells, particularly on SW620 cells, with IC_50_ values ranging from 5.6 µM to 33.0 µM. Benzothiazoles with benzoyl **38b** and picolyl **39c** moiety also exhibited moderate inhibitory potency on additional tumor cell lines. Among amidino benzimidazoles, 1-benzyl-1,2,3-triazole analogs **45a** and **45b** showed the best antiproliferative activity on both sets of tumor cell lines. Of all tested compounds, benzimidazole **45a** with the unsubstituted phenoxy unit showed the most pronounced cell growth inhibition efficiency on THP1 cells in nanomolar range (IC_50_ = 0.8 µM; SI = 69.5; Appendix A). Its fluoro-substituted structural analog **45b** had also marked activity on THP1 cell line with IC_50_ = 2.5µM and SI = 32.4, while methoxy-substituted analog **45c** demonstrated only moderate activity (IC_50_ = 25.1µM; SI > 44) on HL60 cells (Appendix A).

To assess whether cell cycle disturbance is a possible mechanism of action for the antiproliferative activity of the tested compounds, six analogs (**36c**, **42c**, **45a**–**45c**, and **46c**) were tested in cell cycle perturbation experiments on HuT78 cells. As shown in Figure 3a, 24 h post-treatment effects of all applied compounds (5 µM) induced disturbance of cell cycle division in treated compared to control (untreated) cells. Enrichment of the G_0_/G_1_ cell fraction was evident in cell lines treated with all tested analogs, whereas decrease in G_2_/M phase was noticed only following **42c**, **45a**, **45b**, and **45c** cell treatment. Untreated cells followed normal diploid distribution exhibiting regular proliferative features [32].

In our experimental conditions, after 48 h of treatment of HuT78 cells, all tested compounds expressed similar effect pattern. A significant cell cycle perturbation characterized by cell accumulation in the G_2_/M and subG0/G1 phase was evident in treated compared to the non-treated cells (Figure 3b). A SubG1 peak is indicative of DNA fragmentation, plausibly due to apoptosis-induced cell. A decrease in the polyploid cell number of treated compared to the untreated cells was also observed. Polyploidy is a common tumor feature, and pan-cancer analyses confirm that 28.2–37% of human cancers undergo polyploidization [33,34]. Polyploidy enables large phenotypic leaps, providing tumors with access to many different therapy-resistant phenotypes.

Since apoptosis is frequently accompanied by complex mitochondrial changes, alterations in the mitochondrial membrane potential may signal early apoptotic events or, may reflect changes in the apoptotic signaling pathways [35]. In response to multiple intracellular stress conditions, mitochondrial membranes can become permeabilized due to the pore-forming activity of proapoptotic Bcl-2 protein family members. Alternatively, mitochondria can lose their structural integrity after the mitochondrial permeability transition, a phenomenon that is initiated at the mitochondrial inner membrane [36]. In both cases, permeabilized mitochondria allow the release of proapoptotic proteins into the cellular cytoplasm.

To prove apoptosis as a mechanism of treated cells death, we used two methods for apoptosis detection, tracing signs of phosphatildylserine translocation to the extracellular membrane and accompanying changes in the mitochondrial membrane potential (∆Ψm). Results of Annexin-V flow cytometry measurements in selected cell lines, 24 h after compound exposure, showed no significant increase in the proportion of apoptotic cells in treated versus nontreated cells. However, the number of apoptotic cells increased (Figure 4a,b) following 48 h post-treatment with **36c**, **42c**, **45a**–**45c** and **46c**. A statistically significant difference was observed in comparison to cells treated with **42c**, **45a** and **45b** (Figure 4b).

To determine the effects of selected compounds on the function of mitochondria, ΔΨm was assessed in HuT78 cells after 48 h of treatment by chosen compounds (**36c**, **42c**, **45a**–**45c** and **46c**). Changes in the (∆Ψm) were measured using TMRE (Tetramethylrhodamine, Ethyl Ester, Perchlorate) dye. Flow cytometric analysis showed statistically significant changes in mitochondrial membrane potential. Obtained results are consistent with the results of previously published studies which showed that some benzothiazole derivatives have the potential to induce apoptosis of B and T lymphoma cells by the intrinsic pathway through disruption of mitochondrial membranes [8,37].

The given results suggest that disruption of mitochondrial membrane potential produced by tested compounds can lead to cytotoxicity and cell death by apoptosis and/or necrosis as shown in Figure 5.

### 2.3. QSAR Study

Cytotoxic effects of 77 benzothiazoles and benzimidazoles against normal MDCK-1 cells were ranked by activities, and 17 compounds were chosen for the test set. The best QSAR model expressed by multiple linear regression equation generated by five molecular descriptors is:log IC_50_ = −13.88 − 8.18 *SIC1* − 2.27 *GATS4p* − 2.54 *BEHv6* + 14.87 *BELp1* + 6.26 *R7m*(1a)*N*_train_ = 60; *N*_test_ = 17.

Williams plot (Figure 6), which shows the applicability domain of model (1a) detected molecule **25b** as a border outlier and no molecule out of the warning leverage. Outlying behavior of compound **25b** was expected since it demonstrated lowest antiproliferative activity against MDCK-1 cells among two fluoro-substituted benzothiazoles (**16b**, **19b**, **22b**) (Table 1). Removing molecule **25b** from the training set, and subsequent re-analysis produced a following improved QSAR model: log IC_50_ = −14.56 − 8.45 *SIC1* − 2.37 *GATS4p* − 2.70 *BEHv6* + 15.61 *BELp1* + 5.93 *R7m*(1b)*N*_train_ = 59; *N*_test_ = 17.

Experimental and calculated logIC_50_ by Equation (1a,b), as well as values of included descriptors are shown in the Appendix A.

QSAR study for antiproliferative activity on T-cell lymphoma (HuT78) cells was performed on a total of 59 molecules. The set was split into 12 molecules in the test set by activity ranking method, and the remaining 47 candidates were part of the training set. Considering the number of molecules in the dataset, the number of descriptors in the model was limited to four.
Log IC_50_ = 5.14 − 3.38 *MATS8v* − 2.44 *Mor30m* + 1.63 *Mor09p* − 4.12 *E2u*(2a)*N*_train_ = 47; *N*_test_ = 12.

Experimental and calculated logIC_50_ for HuT78 cell line by Equation (2a), as well as values of included descriptors are shown in the Appendix A. Inspection of Williams plot (Figure 7) for the applicability domain of model (2a) revealed three outliers (**25b**, **39c**, **47b**), which have been removed subsequently from the original set. Molecule **25b** had a lower calculated value of logIC_50_ than the measured value, probably because of the presence of two fluorine atoms at the positions R_1_ and R_2_ (Table 1), while molecules **39c** and **47b** exhibited higher activity than calculated. Performing the QSAR analysis on dataset without these outliers, with the same descriptors, resulted on the model of better quality:log IC_50_ = 5.9 − 2.95 *MATS8v* − 2.51 *Mor30m* + 1.79 *Mor09p* − 4.98 *E2u*(2b)*N*_train_ = 45; *N*_test_ = 11.

Calculated log IC_50_ by Equation (2a,b), as well as values of included descriptors are shown in the Appendix A. The statistical results of QSAR models (1a, 1b, 2a, and 2b) are presented in Table 4. Low collinearity among descriptors in all the models was confirmed by a low value of the global correlation among descriptors (*KXX*) and the difference between correlation among the descriptors and response variable (*KXY*) and *KXX* (Δ*K*) higher than 0.05 [38]. The absence of collinearity in models 1a and 2a was also verified by the values of the correlation coefficient (*R* ≤ 0.7) in the correlation matrix (Table 5 and Table 6, respectively).

Statistical parameters, presented in Table 4, confirmed that all four models satisfied fitting abilities: coefficients of determination (*R*^2^*_tr_*) were greater than 0.60, and higher than adjusted coefficient of determination (*R*^2^_adj_). The concordance correlation coefficient of the training set (*CCC*_tr_) was also higher than 0.80 [39]. In order to assess the internal prediction power and stability of QSAR models, leave-one-out (LOO) cross validation technique was performed. The statistical significance of the models was proven by the cross-validated correlation coefficient (*Q*^2^_LOO_), which was higher than 0.05 for all models, but the largest was for model (2b). The differences between *R*^2^ and *Q*^2^_LOO_ did not exceed 0.2–0.3. Additionally, the root-mean-square errors of the cross-validated method (*RMSE_cv_*) were higher than root-mean-square error of the training set (*RMSE_tr_*). An average value of squared correlation coefficients (*r^2^_m_*) between the observed and LOO predicted values of the compounds is a measure for internal validation. Their values were > 0.5 for all models, except for the model (1a). Similarly, the absolute differences between the observed and leave-one-out predicted values of the compounds (Δ*r^2^_m_*) were < 0.2 for all models, except for model (1a), which indicates the low predictive ability of the model (1a), despite of its large *Q*^2^_LOO_ (0.64) [40]. After the exclusion of the outlier, molecule **25b**, the model (1b) exhibited better internal predictivity. Y-Randomization test was performed to check the robustness of the obtained QSAR models. The values of both coefficients, *R*^2^*_Yscr_* (Y-scramble correlation coefficients) and *Q*^2^*_Yscr_* (Y-scramble cross-validation coefficients) were <0.02, implying that models were not obtained by chance [38]. Predictive power of obtained QSAR models were validated by parameters of the external validation, i.e., the coefficients of determination of validation set (*R*^2^*_ext_*) were > 0.60, concordance correlation coefficient of the test set *(CCC_ext_)* was ≥ 0.85 (except for model (1a)), and the root-mean-square error of the external validation set (*RMSE_ex_*), and mean absolute error of the external validation set (*MAE*_ex_) were close to zero. The external performance of all four models in terms of external explained variance (*Q^2^_F1_, Q^2^_F2_, Q^2^_F3_*), which should be > 0.60 was satisfying [41,42]. The chemical domain of applicability defined the structural, physicochemical, and response space of the obtained models. Williams plots (Figure 1 and Figure 2), except mentioned outliers, did not detect structurally influential chemicals in models, in which leverage in the original variable space (*h*) was not higher than warning leverage (*h**) [43]. Generally, QSAR model (2b) for antiproliferative activity on HuT78 cell provided better statistical quality with better predictability in comparation to the model for MDCK-1 cells (1a).

QSAR model for cytotoxic effects against non-tumor MDCK-1 cells contained two BCUT (Burden eigenvalues) descriptors; *BELp1* was the lowest negative eigenvalue num. 1 weighted by polarizability, while the *BEHv6* was the positive highest eigenvalue n. 6 of Burden matrix weighted by atomic van der Waals volumes [44]. Amidino benzothiazoles and amidino benzimidazoles had higher values of *BEHv6* than their halogen- and unsubstituted benzothiazole derivatives (Appendix A), and thus, according to the negative values of *BEHv6* coefficient in the model (1a), lower values of logIC_50_, meaning that they were more toxic against MDCK-1 cells. For example, 6-imidazolyl benzothiazoles had larger substituents at the position R_1_ than their unsubstituted benzothiazole analogs, and therefore had higher values of *BEHv6* and stronger inhibition against MDCK-1 cells (**17a,**
*BEHv6* = 2.173; logIC_50_ = 2.00, and **34a**, *BEHv6* = 3.085; logIC_50_ = 0.45; **20a**, *BEHv6* = 2.719; logIC_50_ = 1.82, and **35a**, *BEHv6* = 2.934; logIC_50_ = 0.43; **23a**, *BEHv6* = 2.711; logIC_50_ = 1.61, and **36a**, *BEHv6* = 2.766; logIC_50_ = 1.50). Values of descriptor *BEHv6* also explain the decrease in cytotoxicity against normal MDCK1 cells by replacement of 1-benzyl-*1H*-1,2,3-triazole moiety in **37a** (*BEHv6* = 3.127; logIC_50_ = 0.18) by substituent of lower total atomic van der Waals volumes, 1*H*-1,2,3-triazole in **36a** (*BEHv6* = 2.766; logIC_50_ = 1.5). Descriptor *BELp1* discriminates well benzothiazoles from benzimidazoles analog, while it was not sensitive to changes of substituents at phenoxymethylene unit within the groups of **a**–**c** compounds (Appendix A). The presence of sulphur atom in benzothiazole, instead of a nitrogen atom in benzimidazole, determined their lower values of *BELp1* descriptors (Appendix A), and thus lower values of logIC_50_ (stronger antiproliferative effect on MDCK-1 cells), which is in accordance with the positive coefficient of these descriptors in models (1a) and (1b). Thus, benzothiazole **41a** had logIC_50_ of 1.76 (*BELp1* = 2.02), while its benzimidazole analog **45a** had higher value of logIC_50_ (logIC_50_ = 2.00, *BELp1* = 2.039). Similarly, benzimidazole **42a** exhibited logIC_50_ of 2.00 (*BELp1* 2.039) that is higher to that of its benzothiazole analog **38a** (logIC_50_ = 1.43, *BELp1* 2.02). R-GETAWAY (Geometry, Topology, and Atom-Weights AssemblY) descriptor *R7m*, encoded the information about the 3D distribution of atomic mass at the topological distance 7 [45,46]. Introduction of the pyrimidinyl group at 6-position of amidino benzothiazoles (**34a**, **34b**, **35c**, **37a**–**37c**) enhanced the values of the *R7m* descriptor compared to their 6-imidazolyl analogs (**38a**, **38b**, **39c**, **41a**–**41c**) (Appendix A). Therefore, in accordance with the positive coefficient of *R7m* in Equation (1a,b), 6-pyrimidinyl benzothiazoles were found to be less toxic on the MDCK-1 cells compared to their 6-imidazolyl analogs. Descriptor *R7m* was extremely sensitive to the difference in the 3D distribution of atomic mass at the topological distance 7 between the benzothiazoles without 1,2,3-triazole moiety. Benzothiazoles with the picolyl aromatic unit (**18a**–**18c**, **19a**–**19c**, **20a**–**20c**) had lower values of *R7m*, and therefore decreased values of logIC_50_ (more active against MDCK-1 cells) compared to their analogs with benzoyl moiety (**15a**–**15c**, **16a**–**16c**, **17a**–**17c**). Descriptor *GATS4p* is a Geary 2D autocorrelations descriptor that reflects a level of independence of polarizability of one atom in the molecular structure on the polarizability of other atoms at the spatial lag 4 [46]. Highest values of these descriptors had 6-fluorobenzothiazoles (**16a**–**16c**, **19a**–**19c**, **22a**–**22c**), since they possess highly polarizable sulphur atom at the topological distance 4 from strongly electrophilic fluorine atom. These compounds were found to be particularly toxic on MDCK-1 cells, which is in accordance with the negative coefficient of *GATS4p* in Equations (1) and (1b). Descriptor *SIC1* is structural information content with the first order of symmetry neighborhood of vertices in a hydrogen-filled graph [47]. Molecules containing three adjacent nitrogen atoms in 1,2,3-triazole moiety had enhanced values of *SIC1* descriptor (Appendix A). This is especially expressed in benzothiazoles with terminal 1*H*-1,2,3-triazole ring (**21a**–**21c**, **22a**–**22c**, **23a**–**23c**), which strongly inhibited MDCK-1 cells.

QSAR models (2a and 2b) for the antiproliferative activities against HuT78 cell contain two 3D-MoRSE (Molecular Representation of Structures based on Electronic diffraction) descriptors, *Mor30m* and *Mor09p*. Descriptors *Mor30m* and *Mor09p* reflect the contribution of the 3D distribution of atomic mass at a scattering parameter *s* = 29 Å^−1^, and atomic polarizability at the scattering parameter *s* = 8 Å^−1^ [48]. The negative coefficient of the descriptor *Mor30m* in models (2a) and (2b) showed that its higher values correspond to a higher antitumor effect (Appendix A). Because of the presence of sulphur atom, amidino benzothiazoles (**34a**–**38a**, **34b**, **36b**–**38b**, **40b**, **34c**–**41c**) had higher values of *Mor30m* than amidino benzimidazoles, which implies lower values of logIC_50_, therefore stronger antiproliferative activity on HuT78 cells. Among amidino benzimidazoles, 1,4-disubstited 1,2,3-triazoles (**45a**–**45c**, **46c**) showed to be the most active (Table 2). These compounds had higher values of *Mor30m* than their unsubstituted 1,2,3-triazole analogs (**44a**–**44c**) (Appendix A) that caused stronger antiproliferative activity on HuT78 cells. The descriptor *Mor09p* is sensitive to the position of atoms with higher polarizability. For example, compound **36c** had higher negative value of *Mor09p* (−1.518, Appendix A) and thus lower value of logIC_50_ (0.2) than compound **36b** with fluorine at the position R_2_ (*Mor09p* = -1.488, logIC_50_ = 0.26). Therefore, substituents with less polarizability decreased the activity against the HuT78 cells. Descriptor *MATS8v* is Moran autocorrelation of lag 8/weighted by atomic van der Waals volumes [48]. This autocorrelation descriptor represents atomic van der Waals volumes at the topological distances 8. Among benzothiazoles **21a**–**21c**, **22a**–**22c** and **23a**–**23c** with terminal 1*H*-1,2,3-triazole ring, compounds with the methoxy group (**21c**, **22c** and **23c**) had higher values of *MATS8v*. The oxygen atom from the methoxy group is at the topological distances 8 from the atom at the R_1_ position. Since oxygen atoms had higher van der Waals volumes than hydrogen or fluorine atoms, these compounds had higher values of *MATS8v* than compounds **21a**, **21b**, **22a**, **22b**, **23a**, **23b** and lowest activity against HuT78. Descriptor *E2u* is 2nd component accessibility directional WHIM (Weighted Holistic Invariant Molecular descriptors) index/unweighted. E is distribution embedded along axes, and it is also directional [48]. Descriptor *E2u* represents a dispersion measure of the projected atoms along the second principal axis, accounting for the molecular size along this principal direction. The compounds **21a**–**21c**, **22a**–**22c** and **23a**–**23c** with terminal 1*H*-1,2,3-triazole ring had higher values of *E2u* descriptor than 1-benzyl-1,2,3-triazoles **24a**–**24c**, **25a**–**25c** and **26a**–**26c,** and therefore exhibited higher activity against HuT78.

## 3. Materials and Methods

### 3.1. General

All the solvents and chemicals were purchased from commercial suppliers Aldrich (St. Louis, MO, USA) and Acros (Geel, Belgium). For monitoring the progress of a reaction and for comparison purpose, thin-layer chromatography (TLC) was performed on precoated Merck silica gel 60F-254 plates (Merck, Kenilworth, NJ, USA) using an appropriate solvent system, and the spots were detected under ultraviolet (UV) light (254 nm). For column chromatography, 0.063–0.2 mm silica gel (Fluka, Seelze, Germany) was employed, glass column was slurry-packed under gravity. Nuclear magnetic resonance (NMR) spectroscopic data for ^1^H and ^13^C nuclei (Appendix A) were recorded at room temperature on a spectrometer Bruker Avance (Bruker, Billerca, MA, USA) 300 MHz and 600 MHz. All NMR spectra were measured in deuterated dimethyl sulfoxide, DMSO with tetramethylsilane as an internal standard. Individual resonances were assigned on the basis of their chemical shifts, signal intensities, multiplicity of resonances, and H–H coupling constants. Melting points were recorded using Kofler micro hot-stage (Reichert, Wien, Austria) and Thermovar HT1BT1 (Reichert, Wien, Austria). Elemental analyses for carbon, hydrogen, and nitrogen were performed on a Perkin–Elmer 2400 elemental analyzer. Analyses are indicated as symbols of elements, and the analytical results obtained are within 0.4% of the theoretical value.

### 3.2. Experimental Procedure for Preparation of Compounds

Compounds 6-chlorobenzothiazol-2-amine **3** [49], 6-fluorobenzothiazol-2-amine **4** [50], 2-amino-5-chlorobenzenethiol **5** [51], 2-amino-5-fluorobenzenethiol **6** [52], **27a** [53], **27b** [54], **27c** [55], **28a** [54], **28b** [54], **28c** [54], **29a** [55], **29b** [54], **29c** [54], **30a** [54], **30b** [28], **30c** [28], **31a** [25], **31b** [54], **31c** [54], **32** [29], **33** [30], hydrochloride **38a** [28], hydrochloride **38b** [28], hydrochloride **39c** [28], hydrochloride **40a** [28], hydrochloride **40b** [28], hydrochloride **40c** [28], hydrochloride **41a** [28], hydrochloride **41c** [28], were synthesized in accordance with procedures given in the literature.

#### 3.2.1. General Procedure for Preparation of 2-(4-Hydroxyphenyl)benzothiazole Derivatives **9a**–**9c**, **10a**–**10c**, and **11a**–**11c**

To a solution of the corresponding 2-aminobenzenethiole (**5**–**7**, 1 eq) in DMF corresponding benzaldehyde (**8a**–**8c**, 1.1 eq) and Na_2_S_2_O_5_ (1.1 eq) were added and reaction mixture was stirred at 100 °C for 2 h. Solvent was evaporated and the residue was purified by column chromatography (CH_2_Cl_2:_CH_3_OH = 50:1).

6-Chloro-2-(4-hydroxyphenyl)benzothiazole **9a.** Compound **9a** was prepared using the above-mentioned procedure from **5** (1.60 g, 10 mmol) and **8a** (1.34 g, 11 mmol) to obtain **9a** as brown powder (1.66 g, 63%; m.p. 169–172 °C). ^1^H NMR (600 MHz, DMSO) δ 10.27 (1H, s, OH), 8.24 (1H, d, *J* = 2.1 Hz), 7.96 (1H, d, *J* = 8.7 Hz), 7.92 (2H, d, *J* = 8.7 Hz), 7.52 (1H, dd, *J* = 8.6, 2.2 Hz), 6.93 (2H, d, *J* = 8.7 Hz). ^13^C NMR (151 MHz, DMSO) δ 168.45, 160.73, 152.48, 135.64, 129.21, 129.12, 126.84, 123.64, 123.35, 121.78, 116.11.

6-Chloro-2-(3-fluoro-4-hydroxyphenyl)benzothiazole **9b.** Compound **9b** was prepared using the above-mentioned procedure from **5** (1.60 mg, 10 mmol) and **8b** (1.54 mg, 11 mmol) to obtain **9b** as brown powder (1.74 g, 62%; m.p. 183–186 °C). ^1^H NMR (600 MHz, DMSO) δ 10.76 (1H, s, OH), 8.26 (1H, d, *J* = 2.1 Hz), 7.98 (1H, d, *J* = 8.7 Hz), 7.85 (1H, dd, *J* = 11.9, 2.1 Hz), 7.74 (1H, dd, *J* = 8.4, 1.9 Hz), 7.54 (1H, dd, *J* = 8.7, 2.2 Hz), 7.13 (1H, t, *J* = 8.6 Hz). ^13^C NMR (75 MHz, DMSO) δ 167.73 (d, *J_CF_* = 2.7 Hz), 151.57 (d, *J_CF_* =241,5 Hz), 152,75, 148.87 (d, *J_CF_* = 12.1 Hz), 136.31, 130.05, 127.50, 124.97 (d, *J_CF_* = 2.8 Hz), 124.54 (d, *J_CF_* = 6.6 Hz), 124.08, 122.38, 118.84 (d, *J_CF_* = 3.3 Hz), 115.20 (d, *J_CF_* = 20.1 Hz).

6-Chloro-2-(4-hydroxy-3-methoxyphenyl)benzothiazole **9c.** Compound **9c** was prepared using the above-mentioned procedure from **5** (1.60 mg, 10 mmol) and **8c** (1.67 mg, 11 mmol) to obtain **9c** as beige powder (1.14 g, 39%; m.p. 219–222 °C). ^1^H NMR (300 MHz, DMSO) δ 9.90 (1H, s, OH), 8.25 (1H, d, *J* = 2.1 Hz), 7.99 (1H, d, *J* = 8.7 Hz), 7.61 (1H, d, *J* = 1.9 Hz), 7.59–7.47 (2H, m), 6.95 (1H, d, *J* = 8.2 Hz), 3.90 (3H, s, OCH_3_). ^13^C NMR (151 MHz, DMSO) δ 168.52, 152.40, 150.27, 148.09, 135.71, 129.26, 126.85, 123.91, 123.37, 121.76, 121.39, 115.91, 110.07, 55.68 (OCH_3_).

6-Fluoro-2-(4-hydroxyphenyl)benzothiazole **10a.** Compound **10a** was prepared using the above-mentioned procedure from **6** (1.43 g, 10 mmol) and **8a** (1.34 g, 11 mmol) to obtain **10a** as white powder (1.84 g, 75%; m.p. 203–205 °C). ^1^H NMR (300 MHz, DMSO) δ 10.24 (1H, s, OH), 8.05–7.95 (2H, m), 7.91 (2H, d, *J* = 8.7 Hz), 7.36 (1H, td, *J* = 9.1, 2.7 Hz), 6.93 (2H, d, *J* = 8.7 Hz). ^13^C NMR (75 MHz, DMSO) δ 168.05 (d, *J_CF_* = 3.2 Hz), 161.00, 159.95 (d, *J_CF_* = 242.3 Hz), 151.02, 135.80 (d, *J_CF_* = 11.6 Hz), 129.45, 124.31, 123.88 (d, *J_CF_* = 9.4 Hz), 116.57, 115.21 (d, *J_CF_* = 24.6 Hz), 109.05 (d, *J_CF_* = 27.2 Hz).

6-Fluoro-2-(3-fluoro-4-hydroxyphenyl)benzothiazole **10b.** Compound **10b** was prepared using the above-mentioned procedure from **6** (1.43 g, 10 mmol) and **8b** (1.54 g, 11 mmol) to obtain **10b** as beige powder (1.02 g, 39%; m.p. 182–185 °C). ^1^H NMR (300 MHz, DMSO) δ 10.74 (1H, s, OH), 8.03 (2H, dd, *J* = 9.0, 6.4, 3.8 Hz), 7.84 (1H, dd, *J* = 11.9, 2.1 Hz), 7.73 (1H, dd, *J* = 8.4, 1.5 Hz), 7.39 (1H, td, *J* = 9.1, 2.7 Hz), 7.13 (1H, t, *J* = 8.7 Hz). ^13^C NMR (75 MHz, DMSO) δ 160.11 (d, *J_CF_* = 242.8 Hz), 151.56 (d, *J_CF_* = 242.8 Hz), 150.83 (d, *J_CF_* = 1.4 Hz), 148.64 (d, *J_CF_* = 12.2 Hz), 135.93, 124.81 (d, *J_CF_* = 2.9 Hz), 124.68, 124.16 (d, *J_CF_* = 9.5 Hz), 118.83 (d, *J_CF_* = 3.3 Hz), 115.44 (d, *J_CF_* = 24.8 Hz), 115.07 (d, *J_CF_* = 20.2 Hz), 109.14 (d, *J_CF_* = 27.4 Hz).

6-Fluoro-2-(4-hydroxy-3-methoxyphenyl)benzothiazole **10c.** Compound **10c** was prepared using the above-mentioned procedure from **6** (1.43 g, 10 mmol) and **8c** (1.67 g, 11 mmol) to obtain **10c** as beige powder (0.93 g, 33%; m.p. 206–208 °C). ^1^H NMR (600 MHz, DMSO) δ 9.86 (1H, s, OH), 8.02 (2H, dd, *J* = 8.7, 3.5 Hz,), 7.61 (1H, d, *J* = 1.8 Hz), 7.49 (1H, dd, *J* = 8.2, 1.9 Hz), 7.37 (1H, td, *J* = 9.1, 2.6 Hz), 6.95 (1H, d, *J* = 8.2 Hz), 3.90 (3H, s, OCH_3_). ^13^C NMR (151 MHz, DMSO) δ 167.64 (d, *J_CF_* = 3.1 Hz), 159.48 (d, *J_CF_* = 242.3 Hz), 150.46, 150.05, 148.09, 135.38 (d, *J_CF_* = 11.8 Hz), 124.09, 123.41 (d, *J_CF_* = 9.5 Hz), 121.22, 115.89, 114.74 (d, *J_CF_* = 24.7 Hz), 109.95, 108.53 (d, *J_CF_* = 27.3 Hz), 55.67 (OCH_3_).

2-(4-Hydroxyphenyl)benzothiazole **11a.** Compound **11a** was prepared using the above-mentioned procedure from **7** (1.25 g, 10 mmol) and **8a** (1.34 g, 11 mmol) to obtain **11a** as beige powder (1.79 g, 78%; m.p. 224–227 °C). ^1^H NMR (300 MHz, DMSO) δ 10.25 (1H, s, OH), 8.08 (1H, d, *J* = 8.0 Hz), 8.00 (1H, d, *J* = 8.1 Hz), 7.96 (2H, d, *J* = 8.7 Hz), 7.51 (1H, t, *J* = 7.7 Hz), 7.41 (1H, t, *J* = 7.6 Hz), 6.97 (2H, d, *J* = 8.7 Hz). ^13^C NMR (75 MHz, DMSO) δ 167.92, 160.99, 154.20, 134.58, 129.51, 126.86, 125.34, 124.52, 122.76, 122.54, 116.55.

2-(3-Fluoro-4-hydroxyphenyl)benzothiazole **11b.** Compound **11b** was prepared using the above-mentioned procedure from **7** (1.25 g, 10 mmol) and **8b** (1.54 g, 11 mmol) to obtain **11b** as yellow powder (1.92 g, 78%; m.p. 199–201 °C). ^1^H NMR (300 MHz, DMSO) δ 10.72 (1H, s, OH), 8.11 (1H, d, *J* = 7.5 Hz), 8.02 (1H, d, *J* = 7.8 Hz), 7.87 (1H, dd, *J* = 12.0, 2.1 Hz), 7.76 (1H, dd, *J* = 8.4, 1.4 Hz), 7.53 (1H, t, *J* = 8.2 Hz), 7.44 (1H, t, *J* = 8.1 Hz), 7.14 (1H, t, *J* = 8.7 Hz). ^13^C NMR (75 MHz, DMSO) δ 166.68, 153.98, 151.57 (d, *J_CF_* = 242.8 Hz), 148.61 (d, *J_CF_* = 12.1 Hz), 134.76, 127.03, 125.67, 125.00, 124.84 (d, *J_CF_* = 2.9 Hz), 122.98, 122.67, 118.81 (d, *J_CF_* = 3.3 Hz), 115.12 (d, *J_CF_* = 20.1 Hz).

2-(4-Hydroxy-3-methoxyphenyl)benzothiazole **11c.** Compound **11c** was prepared using the above-mentioned procedure from **7** (1.25 g, 10 mmol) and **8c** (1.67 g, 11 mmol) to obtain **11c** as a beige powder (2.19 g, 85%; m.p. 185–187 °C). ^1^H NMR (300 MHz, DMSO) δ 9.86 (1H, s, OH), 8.08 (1H, d, *J* = 7.4 Hz), 8.01 (1H, d, *J* = 7.7 Hz), 7.64 (1H, d, *J* = 2.0 Hz), 7.59–7.47 (2H, m), 7.41 (1H, t, *J* = 8.1 Hz), 6.96 (1H, d, *J* = 8.2 Hz), 3.91 (3H, s, OCH_3_). ^13^C NMR (151 MHz, DMSO) δ 167.50, 153.61, 150.01, 148.06, 134.14, 126.40, 124.89, 124.30, 122.27, 122.05, 121.25, 115.87, 110.05, 55.67 (OCH_3_).

#### 3.2.2. General Procedure for O-Alkylation of Propargylated Benzothiazole Derivatives **12a**–**12c**, **13a**–**13c**, **14a**–**14c** and Target Analogs **15a**–**15c**, **16a**–**16c**, **17a**–**17c**, **18a**–**18c**, **19a**–**19c**, **20a**–**20c**

To a solution of the corresponding heterocyclic base (**9a**–**9c, 10a**–**10c, 11a**–**11c**; 1 eq) in acetonitrile, K_2_CO_3_ (3 eq) was added and stirred for 30 min. Corresponding alkyl halogenide (1.2 eq) was added and the reaction mixture was stirred for 12 h at room temperature. The solvent was evaporated and the residue was purified by column chromatography (CH_2_Cl_2_:CH_3_OH = 50:1).

6-Chloro-2-(4-(prop-2-yn-1-yloxy)phenyl)benzothiazole **12a.** Using the above-mentioned procedure from **9a** (500 mg, 1.91 mmol) and propargyl bromide (174 µL, 2.29 mmol), compound **12a** was obtained as beige powder (471.8 mg, 82%; m.p. 158–161 °C). ^1^H NMR (300 MHz, DMSO) δ 8.29 (1H, d, *J* = 2.1 Hz), 8.05 (2H, d, *J* = 8.9 Hz), 8.01 (1H, d, *J* = 8.7 Hz), 7.55 (1H, dd, *J* = 8.7, 2.2 Hz), 7.18 (2H, d, *J* = 8.9 Hz), 4.93 (2H, d, *J* = 2.3 Hz, OCH_2_), 3.65 (1H, t, *J* = 2.3 Hz, CH). ^13^C NMR (151 MHz, DMSO) δ 167.91, 159.83, 152.40, 135.83, 129.53, 128.89, 126.98, 125.80, 123.62, 121.90, 115.66, 78.74(OCH_2_CCH), 78.70(OCH_2_CCH), 55.71 (OCH_2_CCH).

6-Chloro-2-(3-fluoro-4-(prop-2-yn-1-yloxy)phenyl)benzothiazole **12b.** Using the above-mentioned procedure from **9b** (500 mg, 1.79 mmol) and propargyl bromide (163 µL, 2.15 mmol), compound **12b** was obtained as beige powder (500.5 mg, 88%; m.p. 142–145 °C). ^1^H NMR (300 MHz, DMSO) δ 8.32 (1H, d, *J* = 2.2 Hz), 8.03 (1H, d, *J* = 8.7 Hz), 7.99–7.87 (3H, m), 7.57 (1H, dd, *J* = 8.7, 2.2 Hz), 7.43 (1H, t, *J* = 8.5 Hz), 5.03 (2H, d, *J* = 2.3 Hz, OCH_2_), 3.72 (1H, t, *J* = 2.3 Hz, CH). ^13^C NMR (151 MHz, DMSO) δ 166.73, 152.19, 151.76 (d, *J_CF_* = 246.2 Hz), 147.68 (d, *J_CF_* = 10.7 Hz), 136.02, 129.87, 127.15, 126.27 (d, *J_CF_* = 7.0 Hz), 124.24 (d, *J_CF_* = 3.0 Hz), 123.84, 122.01, 115.93, 114.54 (d, *J_CF_* = 20.2 Hz), 79.34 (OCH_2_CCH), 78.24 (OCH_2_CCH), 56.67 (OCH_2_CCH).

6-Chloro-2-(3-methoxy-4-(prop-2-yn-1-yloxy)phenyl)benzothiazole **12c.** Using the above-mentioned procedure from **9c** (500 mg, 1.71 mmol) and propargyl bromide (156 µL, 2.05 mmol), compound **12c** was obtained as beige powder (497.4 mg, 88%; m.p. 153–156 °C). ^1^H NMR (300 MHz, DMSO) δ 8.26 (1H, d, *J* = 2.1 Hz), 8.01 (1H, d, *J* = 8.7 Hz), 7.68–7.57 (2H, m), 7.53 (1H, dd, *J* = 8.7, 2.2 Hz), 7.18 (1H, d, *J* = 8.3 Hz), 4.90 (2H, d, *J* = 2.3 Hz, OCH_2_), 3.88 (3H, s, OCH_3_), 3.62 (1H, t, *J* = 2.3 Hz, CH). ^13^C NMR (75 MHz, DMSO) δ 168.58, 152.82, 149.92, 136.39, 130.07, 127.49, 126.55, 124.14, 122.37, 121.18, 114.30, 110.21, 79.25(OCH_2_CCH), 56.55 (OCH_2_CCH), 56.16 (OCH_3_).

6-Fluoro-2-(4-(prop-2-yn-1-yloxy)phenyl)benzothiazole **13a.** Using the above-mentioned procedure from **10a** (500 mg, 2.04 mmol) and propargyl bromide (186 µL, 2.45 mmol), compound **13a** was obtained as beige powder (338.5 mg, 58%; m.p. 132–135 °C). ^1^H NMR (600 MHz, DMSO) δ 8.06–8.00 (4H, m), 7.39 (1H, td, *J* = 9.1, 2.6 Hz), 7.20–7.16 (2H, m), 4.93 (2H, d, *J* = 2.3 Hz, OCH_2_), 3.65 (1H, t, *J* = 2.3 Hz, CH). ^13^C NMR (151 MHz, DMSO) δ 167.00 (d, *J_CF_* = 2.5 Hz), 159.63, 159.60 (d, *J_CF_* = 242.7 Hz), 150.46, 135.51 (d, *J_CF_* = 11.9 Hz), 128.72, 125.96, 123.67 (d, *J_CF_* = 9.5 Hz), 115.61, 114.89 (d, *J_CF_* = 24.9 Hz), 108.63 (d, *J_CF_* = 27.1 Hz), 78.76 (OCH_2_CCH), 78.69 (OCH_2_CCH), 55.68 (OCH_2_CCH).

6-Fluoro-2-(3-fluoro-4-(prop-2-yn-1-yloxy)phenyl)benzothiazole **13b.** Using the above-mentioned procedure from **10b** (500 mg, 1.89 mmol) and propargyl bromide (172 µL, 2.27 mmol), compound **13b** was obtained as beige powder (314.7 mg, 55%; m.p. 125–128 °C). ^1^H NMR (400 MHz, DMSO) δ 8.10–8.02 (2H, m), 7.93 (1H, dd, *J* = 11.9, 2.2 Hz), 7.91–7.86 (1H, m), 7.45–7.38 (2H, m), 5.03 (2H, d, *J* = 2.4 Hz, OCH_2_), 3.72 (1H, t, *J* = 2.4 Hz, CH). ^13^C NMR (101 MHz, DMSO) δ 166.35, 160.26 (d, *J_CF_* = 243.1 Hz), 152.27 (d, *J_CF_* = 246.1 Hz), 150.78, 147.99 (d, *J_CF_* = 10.6 Hz), 136.25 (d, *J_CF_* = 11.9 Hz), 126.93 (d, *J_CF_* = 6.9 Hz), 124.57 (d, *J_CF_* = 2.6 Hz), 124.45 (d, *J_CF_* = 9.6 Hz), 116.42, 115.64 (d, *J_CF_* = 24.8 Hz), 114.92 (d, *J_CF_* = 20.2 Hz), 109.24 (d, *J_CF_* = 27.4 Hz), 79.84 (OCH_2_CCH), 78.77 (OCH_2_CCH), 57.15 (OCH_2_CCH).

6-Fluoro-2-(3-methoxy-4-(prop-2-yn-1-yloxy)phenyl)benzothiazole **13c.** Using the above-mentioned procedure from **10c** (500 mg, 1.82 mmol) and propargyl bromide (166 µL, 2.18 mmol), compound **13c** was obtained as a beige powder (223.0 mg, 39%; m.p. 145–148 °C). ^1^H NMR (300 MHz, DMSO) δ 8.05 (2H, dt, *J* = 8.5, 3.6 Hz), 7.65 (1H, d, *J* = 2.1 Hz), 7.60 (1H, dd, *J* = 8.3, 2.1 Hz), 7.39 (1H, td, *J* = 9.1, 2.7 Hz), 7.20 (1H, d, *J* = 8.4 Hz), 4.92 (2H, d, *J* = 2.4 Hz, OCH_2_), 3.91 (3H, s, OCH_3_), 3.64 (1H, t, *J* = 2.3 Hz, CH). ^13^C NMR (75 MHz, DMSO) δ 167.67 (d, *J_CF_* = 3.2 Hz), 160.13 (d, *J_CF_* = 242.7 Hz), 150.89, 149.93, 149.72, 136.08 (d, *J_CF_* = 11.8 Hz), 126.73, 124.20 (d, *J_CF_* = 9.5 Hz), 121.01, 115.41 (d, *J_CF_* = 24.8 Hz), 114.30, 110.09, 109.11 (d, *J_CF_* = 27.3 Hz), 79.30 (OCH_2_CCH), 79.23 (OCH_2_CCH), 56.55 (OCH_2_CCH), 56.15 (OCH_3_).

2-(4-(Prop-2-yn-1-yloxy)phenyl)benzothiazole **14a.** Using the above-mentioned procedure from **11a** (500 mg, 2.19 mmol) and propargyl bromide (200 µL, 2.63 mmol), compound **14a** was obtained as white powder (517.1 mg, 88%; m.p. 136–140 °C). ^1^H NMR (600 MHz, DMSO) δ 8.12 (1H, d, *J* = 7.8 Hz), 8.06 (2H, d, *J* = 8.8 Hz), 8.02 (1H, d, *J* = 8.1 Hz), 7.53 (1H, t, *J* = 8.2 Hz), 7.44 (1H, t, *J* = 8.1 Hz), 7.18 (2H, d, *J* = 8.9 Hz), 4.93 (2H, d, *J* = 2.3 Hz, OCH_2_), 3.64 (1H, t, *J* = 2.3 Hz, CH). ^13^C NMR (151 MHz, DMSO) δ 166.87, 159.63, 153.62, 134.26, 128.78, 126.52, 126.17, 125.15, 122.51, 122.20, 115.61, 78.78 (OCH_2_CCH), 78.67 (OCH_2_CCH), 55.69 (OCH_2_CCH).

2-(3-Fluoro-4-(prop-2-yn-1-yloxy)phenyl)benzothiazole **14b.** Using the above-mentioned procedure from **11b** (500 mg, 2.04 mmol) and propargyl bromide (186 µL, 2.45 mmol), compound **14b** was obtained as beige powder (439.3 mg, 76%; m.p. 134–138 °C). ^1^H NMR (300 MHz, DMSO) δ 8.17–8.12 (1H, m), 8.07–8.02 (1H, m), 7.99–7.88 (2H, m), 7.58–7.51 (1H, m), 7.50–7.38 (2H, m), 5.03 (2H, d, *J* = 2.4 Hz, OCH_2_), 3.71 (1H, t, *J* = 2.4 Hz, CH). ^13^C NMR (75 MHz, DMSO) δ 153.91, 152.26 (d, *J_CF_* = 245.9 Hz), 147.96 (d, *J_CF_* = 10.6 Hz), 134.96, 127.17, 127.09, 125.97, 124.60 (d, *J_CF_* = 3.3 Hz), 123.21, 122.82, 116.42 (d, *J_CF_* = 1.6 Hz), 114.97 (d, *J_CF_* = 20.2 Hz), 79.82 (OCH_2_CCH), 78.77 (OCH_2_CCH), 57.14 (OCH_2_CCH).

2-(3-Methoxy-4-(prop-2-yn-1-yloxy)phenyl)benzothiazole **14c.** Using the above-mentioned procedure from **11c** (500 mg, 1.94 mmol) and propargyl bromide (177 µL, 2.33 mmol), compound **14c** was obtained as beige powder (397.1 mg, 69%; m.p. 122–125 °C). ^1^H NMR (600 MHz, DMSO) δ 8.11 (1H, d, *J* = 7.8 Hz), 8.04 (1H, d, *J* = 8.1 Hz), 7.68 (1H, d, *J* = 2.1 Hz), 7.63 (1H, dd, *J* = 8.3, 2.1 Hz), 7.53 (1H, t, *J* = 8.2 Hz), 7.44 (1H, t, *J* = 8.1 Hz), 7.21 (1H, d, *J* = 8.4 Hz), 4.91 (2H, d, *J* = 2.4 Hz, OCH_2_), 3.91 (3H, s, OCH_3_), 3.63 (1H, t, *J* = 2.3 Hz, CH). ^13^C NMR (151 MHz, DMSO) δ 167.03, 153.54, 149.45, 149.22, 134.33, 126.52, 126.47, 125.18, 122.52, 122.17, 120.53, 113.86, 109.74, 78.82 (OCH_2_CCH), 78.70 (OCH_2_CCH), 56.07 (OCH_2_CCH), 55.67 (OCH_3_).

6-Chloro-2-(4-(2-oxo-2-phenylethoxy)phenyl)benzothiazole **15a.** Using the above-mentioned procedure from **9a** (80.0 mg, 0.31 mmol) and 2-bromoacetophenone (74.0 mg, 0.37 mmol), compound **15a** was obtained as a grey powder (53.2 mg, 45%; m.p. 192–195 °C). ^1^H NMR (300 MHz. DMSO) δ 8.28 (1H, d, *J* = 2.0 Hz), 8.10–7.96 (5H, m), 7.72 (1H, t, *J* = 7.4 Hz), 7.64–7.51 (3H, m), 7.17 (2H, d, *J* = 8.9 Hz), 5.75 (2H, s, OCH_2_). ^13^C NMR (151 MHz. DMSO) δ 194.03 (C=O), 167.98, 160.71, 152.43, 135.81, 134.24, 133.87, 129.47, 128.86, 128.84, 127.87, 126.96, 125.47, 123.58, 121.88, 115.50, 70.29 (OCH_2_). Anal.calcd. for C_21_H_14_ClNO_2_S (Mr = 379.86): C 66.40, H 3.72, N 3.69; found: C 66.17, H 3.71, N 3.67.

6-Chloro-2-(3-fluoro-4-(2-oxo-2-phenylethoxy)phenyl)benzothiazole **15b.** Using the above-mentioned procedure from **9b** (80.0 mg, 0.29 mmol) and 2-bromoacetophenone (69.3 mg, 0.35 mmol), compound **15b** was obtained as a white powder (43.4 mg, 38%; m.p. 192–195 °C). ^1^H NMR (300 MHz. DMSO) δ 8.31 (1H, d, *J* = 2.1 Hz), 8.08–8.00 (3H, m), 7.96 (1H, dd, *J* = 12.0, 2.1 Hz), 7.82 (1H, d, *J* = 8.6 Hz), 7.73 (1H, t, *J* = 7.4 Hz), 7.65–7.53 (3H, m), 7.32 (1H, t, *J* = 8.7 Hz), 5.87 (2H, s, OCH_2_). ^13^C NMR (151 MHz, DMSO) δ 193.60 (C=O), 166.83, 151.50 (d, *J_CF_* = 245.8 Hz), 152.22, 148.73, 148.67, 136.00, 134.08, 133.96, 129.80, 128.85, 127.87, 127.12, 125.79 (d, *J_CF_* = 6.8 Hz), 124.15 (d, *J_CF_* = 2.9 Hz), 123.79, 121.98, 115.63, 114.55 (d, *J_CF_* = 20.2 Hz), 70.85 (OCH_2_). Anal.calcd. for C_21_H_13_ClFNO_2_S (Mr = 397.85): C 63.40, H 3.29, N 3.52; found: C 63.18, H 3.29, N 3.49.

6-Chloro-2-(3-methoxy-4-(2-oxo-2-phenylethoxy)phenyl)benzothiazole **15c.** Using the above mentioned procedure from **9c** (60.0 mg,  0.21 mmol) and 2-bromoacetophenone (50.2 mg, 0.25 mmol) compound **15c** was obtained as grey powder (65.2 mg, 77%; m.p. 163–166 °C. ^1^H NMR (300 MHz, DMSO) δ 8.27 (1H, d, *J* = 2.0 Hz), 8.04 (3H, t, *J* = 7.6 Hz), 7.77–7.64 (2H, m), 7.64–7.52 (4H, m), 7.07 (1H, d, *J* = 8.5 Hz), 5.74 (2H, s, OCH_2_), 3.94 (3H, s, OCH_3_). ^13^C NMR (75 MHz, DMSO) δ 194.52 (C=O), 168.64, 152.85, 150.91, 149.65, 136.36, 134.74, 134.36, 130.00, 129.33, 128.38, 127.46, 126.11, 124.09, 122.35, 121.23, 113.91, 110.38, 71.07 (OCH_2_), 56.23(OCH_3_). Anal.calcd. for C_22_H_16_ClNO_3_S (Mr = 409.88): C 64.47, H 3.93, N 3.42; found: C 64.24, H 3.93, N 3.40.

6-Fluoro-2-(4-(2-oxo-2-phenylethoxy)phenyl)benzothiazole **16a.** Using the above-mentioned procedure from **10a** (90.0 mg, 0.37 mmol) and 2-bromoacetophenone (88.4 mg, 0.44 mmol), compound **16a** was obtained as a yellow powder (68.7 mg, 51%; m.p. 164–167 °C). ^1^H NMR (600 MHz, DMSO) δ 8.08–7.97 (6H, m), 7.72 (1H, t, *J* = 7.4 Hz), 7.60 (2H, t, *J* = 7.8 Hz), 7.39 (1H, td, *J* = 9.1, 2.7 Hz), 7.16 (2H, d, *J* = 8.8 Hz), 5.74 (2H, s, OCH_2_). ^13^C NMR (151 MHz, DMSO) δ 194.06 (C=O), 167.08 (d, *J_CF_* = 3.0 Hz), 160.52, 159.58 (d, *J_CF_* = 242.7 Hz), 150.49, 135.49 (d, *J_CF_* = 11.8 Hz), 134.24, 133.87, 128.83, 128.70, 127.87, 125.64, 123.64 (d, *J_CF_* = 9.5 Hz), 115.46, 114.87 (d, *J_CF_* = 24.7 Hz), 108.61 (d, *J* _CF_= 27.4 Hz), 70.27 (OCH_2_). Anal.calcd. for C_21_H_14_FNO_2_S (Mr = 363.41): C 69.41, H 3.88, N 3.85; found: C 69.14, H 3.88, N 3.83.

6-Fluoro-2-(3-fluoro-4-(2-oxo-2-phenylethoxy)phenyl)benzothiazole **16b.** Using the above-mentioned procedure from **10b** (90.0 mg, 0.34 mmol) and 2-bromoacetophenone (81.2 mg, 0.41 mmol), compound **16b** was obtained as an orange powder (57.0 mg, 60%; m.p. 148–152 °C). ^1^H NMR (600 MHz, DMSO) δ 8.11–8.00 (H4, m), 7.94 (1H, dd, *J* = 12.0, 1.9 Hz), 7.80 (1H, d, *J* = 8.5 Hz), 7.72 (1H, t, *J* = 7.4 Hz), 7.60 (2H, t, *J* = 7.7 Hz), 7.41 (1H, td, *J* = 9.0, 2.6 Hz), 7.31 (1H, t, *J* = 8.6 Hz), 5.86 (2H, s, OCH_2_). ^13^C NMR (75 MHz, DMSO) δ 194.13 (C=O), 166.41, 160.22 (d, *J_CF_* = 242.9 Hz), 152.00 (d, *J_CF_* = 245.7 Hz), 150.79, 149.01 (d, *J_CF_* = 10.4 Hz), 136.21 (d, *J_CF_* = 11.8 Hz), 134.57, 134.47, 129.35, 128.37, 126.44 (d, *J_CF_* = 6.8 Hz), 124.43 (t, *J_CF_* = 6.4 Hz), 124.40 (d, *J_CF_* = 9.6 Hz), 116.11, 115.60 (d, *J_CF_* = 24.9 Hz), 114.93 (d, *J_CF_* = 20.3 Hz), 109.22 (d, *J_CF_* = 27.4 Hz), 71.32 (OCH_2_). Anal.calcd. for C_21_H_13_F_2_NO_2_S (Mr = 381.40): C 66.13, H 3.44, N 3.67; found: C 65.90, H 3.43, N 3.66.

6-Fluoro-2-(3-methoxy-4-(2-oxo-2-phenylethoxy)phenyl)benzothiazole **16c.** Using the above-mentioned procedure from **10c** (90.0 mg, 0.33 mmol) and 2-bromoacetophenone (78.8 mg, 0.40 mmol), compound **16c** was obtained as a beige powder (53.5 mg, 41%; m.p. 178–181 °C). ^1^H NMR (600 MHz, DMSO) δ 8.08–8.00 (4H, m), 7.71 (1H, t, *J* = 7.4 Hz), 7.66 (1H, d, *J* = 2.0 Hz), 7.59 (2H, t, *J* = 7.8 Hz), 7.53 (1H, dd, *J* = 8.4, 2.0 Hz), 7.39 (1H, td, *J* = 9.0, 2.6 Hz), 7.06 (1H, d, *J* = 8.5 Hz), 5.72 (2H, s, OCH_2_), 3.93 (3H, s, OCH_3_). ^13^C NMR (151 MHz, DMSO) δ 194.07 (C=O), 167.25, 159.60 (d, *J_CF_* = 242.7 Hz), 150.42, 150.24, 149.18, 135.56 (d, *J_CF_* = 11.7 Hz), 134.27, 133.84, 128.82, 127.87, 125.83, 123.67 (d, *J_CF_* = 9.4 Hz), 120.58, 114.89 (d, *J_CF_* = 24.8 Hz), 113.48, 109.86, 108.60 (d, *J_CF_* = 27.3 Hz), 70.60 (OCH_2_), 55.76 (OCH_3_). Anal.calcd. for C_22_H_16_FNO_3_S (Mr = 393.43): C 67.16, H 4.10, N 3.56; found: C 66.91, H 4.09, N 3.54.

2-(4-(2-Oxo-2-phenylethoxy)phenyl)benzothiazole **17a.** Using the above-mentioned procedure from **11a** (120 mg, 0.53 mmol) and 2-bromoacetophenone (126.6 mg, 0.64 mmol), compound **17a** was obtained as a beige powder (64.3 mg, 35%; m.p. 150–155 °C). ^1^H NMR (300 MHz, DMSO) δ 8.13–8.07 (1H, m), 8.07–7.97 (5H, m), 7.75–7.67 (1H, m), 7.62–7.54 (2H, m), 7.54–7.47 (1H, m), 7.45–7.37 (1H, m), 7.15 (2H, d, *J* = 8.9 Hz), 5.73 (2H, s, OCH_2_). ^13^C NMR (75 MHz, DMSO) δ 194.58 (C=O), 167.45, 161.00, 154.13, 134.73, 134.38, 129.34, 129.26, 128.37, 127.00, 126.33, 125.61, 122.96, 122.68, 115.94, 70.77 (OCH_2_). Anal.calcd. for C_21_H_15_NO_2_S (Mr = 345.42): C 73.02, H 4.38, N 4.06; found: C 72.72, H 4.37, N 4.03.

2-(3-Fluoro-4-(2-oxo-2-phenylethoxy)phenyl)benzothiazole **17b.** Using the above-mentioned procedure from **11b** (120 mg, 0.49 mmol) and 2-bromoacetophenone (117.0 mg, 0.59 mmol), compound **17b** was obtained as a beige powder (103.5 mg, 58%; m.p. 174–177 °C). ^1^H NMR (600 MHz, DMSO) δ 8.13 (1H, d, *J* = 7.8 Hz), 8.06–8.02 (3H, m), 7.96 (1H, dd, *J* = 12.0, 2.1 Hz), 7.83–7.80 (1H, m), 7.73 (1H, t, *J* = 7.4 Hz), 7.60 (2H, t, *J* = 7.8 Hz), 7.56–7.52 (1H, m), 7.48–7.43 (1H, m), 7.32 (1H, t, *J* = 8.7 Hz), 5.86 (2H, s, OCH_2_). ^13^C NMR (151 MHz, DMSO) δ 193.64 (C=O), 165.78, 153.44, 151.51 (d, *J_CF_* = 245.6 Hz), 148.49 (d, *J_CF_* = 10.3 Hz), 134.44, 134.10, 133.96, 128.85, 127.87, 126.64, 126.17 (d, *J_CF_* = 6.8 Hz), 125.40, 124.01 (d, *J_CF_* = 2.5 Hz), 122.67, 122.28, 115.60, 114.47 (d, *J* = 20.2 Hz), 70.84 (OCH_2_). Anal.calcd. for C_21_H_14_FNO_2_S (Mr = 363.41): C 69.41, H 3.88, N 3.85; found: C 69.12, H 3.87, N 3.83.

2-(3-Methoxy-4-(2-oxo-2-phenylethoxy)phenyl)benzothiazole **17c.** Using the above-mentioned procedure from **11c** (120 mg, 0.47 mmol) and 2-bromoacetophenone (112.3 mg, 0.56 mmol), compound **17c** was obtained as a beige powder (99.2 mg, 56%; m.p. 184–186 °C). ^1^H NMR (400 MHz, DMSO) δ 8.11 (1H, d, *J* = 7.5 Hz), 8.07–8.02 (3H, m), 7.75–7.68 (2H, m), 7.63–7.50 (4H, m), 7.44 (1H, t, *J* = 7.6 Hz), 7.07 (1H, d, *J* = 8.5 Hz), 5.73 (2H, s, OCH_2_), 3.95 (3H, OCH_3_). ^13^C NMR (101 MHz, DMSO) δ 194.58 (C=O), 167.61, 154.07, 150.71, 149.65, 134.80, 134.77, 134.37, 129.34, 128.39, 127.02, 126.51, 125.65, 122.99, 122.68, 121.11, 113.93, 110.39, 71.09 (OCH_2_), 56.25 (OCH_3_). Anal.calcd. for C_22_H_17_NO_3_S (Mr = 375.44): C 70.38, H 4.56, N 3.73; found: C 70.13, H 4.55, N 3.71.

6-Chloro-2-(4-(pyridin-2-yl)methoxy)phenyl)benzothiazole **18a.** Using the above-mentioned procedure from **9a** (80.0 mg, 0.31 mmol) 2-(bromomethyl)pyridine hydrobromide (94.1 mg, 0.37 mmol), compound **18a** was obtained as a beige powder (72.0 mg, 66%; m.p. 199–204 °C). ^1^H NMR (300 MHz, DMSO) δ 8.60 (1H, d, *J* = 4.1 Hz), 8.27 (1H, d, *J* = 2.1 Hz), 8.04 (2H, d, *J* = 8.9 Hz), 8.00 (1H, d, *J* = 8.7 Hz), 7.86 (1H, td, *J* = 7.7, 1.7 Hz), 7.54 (2H, dd, *J* = 8.7, 2.2 Hz), 7.37 (1H, dd, *J* = 7.5, 4.9 Hz), 7.22 (2H, d, *J* = 8.9 Hz), 5.30 (2H, s, OCH_2_). ^13^C NMR (151 MHz, DMSO) δ 167.96, 160.83, 156.10, 152.42, 149.15, 137.09, 135.81, 129.49, 129.01, 126.97, 125.53, 123.60, 123.11, 121.90, 121.79, 115.64, 70.48 (OCH_2_). Anal.calcd. for C_19_H_13_ClNO_2_S (Mr = 352.84): C 64.68, H 3.71, N 7.94; found: C 64.44, H 3.71, N 7.90.

6-Chloro-2-(3-fluoro-4-(pyridin-2-yl)methoxy)phenyl)benzothiazole **18b.** Using the above-mentioned procedure from **9b** (80.0 mg, 0.29 mmol) 2-(bromomethyl)pyridine hydrobromide (88.0 mg, 0.35 mmol), compound **18b** was obtained as a white powder (43.0 mg, 40%; m.p. 211–215 °C). ^1^H NMR (600 MHz, DMSO) δ 8.57 (1H, ddd, *J* = 4.9, 1.6, 1.0 Hz), 8.22 (1H, d, *J* = 2.1 Hz), 7.98 (1H, d, *J* = 8.7 Hz), 7.89 (1H, dd, *J* = 12.0, 2.2 Hz), 7.86–7.79 (2H, m), 7.55–7.50 (2H, m), 7.41 (1H, t, *J* = 8.6 Hz), 7.34 (1H, ddd, *J* = 7.5, 4.8, 0.8 Hz), 5.35 (2H, s, OCH_2_). ^13^C NMR (151 MHz, DMSO) δ 166.77, 155.71, 151.95 (d, *J_CF_* = 246.4 Hz), 152.30, 149.19, 148.90 (d, *J_CF_* = 10.5 Hz), 137.01, 136.08, 129.92, 127.09, 126.18 (d, *J_CF_* = 6.9 Hz), 124.34 (d, *J_CF_* = 3.2 Hz), 123.80, 123.17, 121.86, 121.83, 116.18, 114.57 (d, *J_CF_* = 20.3 Hz), 71.67 (OCH_2_). Anal.calcd. for C_19_H_12_ClFN_2_OS (Mr = 370.83): C 61.54, H 3.26, N 7.55; found: C 61.31, H 3.25, N 7.52.

6-Chloro-2-(3-methoxy-4-(pyridin-2-yl)methoxy)phenyl)benzothiazole **18c.** Using the above-mentioned procedure from **9c** (60.0 mg, 0.21 mmol) 2-(bromomethyl)pyridine hydrobromide (63.7 mg, 0.25 mmol), compound **18c** was obtained as a grey powder (64.0 mg, 81%; m.p. 173–176 °C). ^1^H NMR (600 MHz, DMSO) δ 8.18 (ddd, *J* = 4.8, 1.5, 0.8 Hz, 1H), 7.85 (d, *J* = 2.2 Hz, 1H), 7.60 (d, *J* = 8.7 Hz, 1H), 7.45 (td, *J* = 7.7, 1.8 Hz, 1H), 7.26 (d, *J* = 2.1 Hz, 1H), 7.18 (dd, *J* = 8.4, 2.1 Hz, 1H), 7.13 (dd, *J* = 8.7, 2.2 Hz, 2H), 6.95 (ddd, *J* = 7.4, 4.8, 0.8 Hz, 1H), 6.80 (d, *J* = 8.5 Hz, 1H), 4.86 (2H, s, OCH_2_), 3.51 (3H, s, OCH_3_). ^13^C NMR (151 MHz, DMSO) δ 168.13, 156.18, 152.33, 150.54, 149.38, 149.14, 137.06, 135.86, 129.54, 126.98, 125.69, 123.60, 123.10, 121.84, 121.80, 120.94, 113.52, 109.74, 70.91 (OCH_2_), 55.75 (OCH_3_). Anal.calcd. for C_20_H_15_ClN_2_O_2_S (Mr = 382.86): C 62.74, H 3.95, N 7.32; found: C 62.51, H 3.94, N 7.28.

6-Fluoro-2-(4-(pyridin-2-yl)methoxy)phenyl)benzothiazole **19a.** Using the above-mentioned procedure from **10a** (70.0 mg, 0.29 mmol) 2-(bromomethyl)pyridine hydrobromide (88.0 mg, 0.35 mmol), compound **19a** was obtained as a beige powder (55.9 mg, 58%; m.p. 165–170 °C). ^1^H NMR (300 MHz, DMSO) δ 8.60 (1H, d, *J* = 4.8 Hz), 8.08–7.97 (4H, m), 7.86 (1H, td, *J* = 7.7, 1.7 Hz), 7.55 (1H, d, *J* = 7.8 Hz), 7.43–7.33 (2H, m), 7.22 (2H, d, *J* = 8.9 Hz), 5.30 (2H, s, OCH_2_). ^13^C NMR (75 MHz, DMSO) δ 161.13, 160.08 (d, *J_CF_* = 242.5 Hz), 156.64, 150.98, 149.67, 137.55, 135.98 (d, *J_CF_* = 11.9 Hz), 129.34, 126.17, 124.15 (d, *J_CF_* = 9.5 Hz), 123.59, 122.26, 116.09, 115.39 (d, *J_CF_* = 24.9 Hz), 109.14 (d, *J_CF_* = 27.3 Hz), 70.97 (OCH_2_). Anal.calcd. for C_19_H_13_FN_2_OS (Mr = 336.38): C 67.84, H 3.90, N 8.33; found: C 67.57, H 3.90, N 8.29.

6-Fluoro-2-(3-fluoro-4-(pyridin-2-yl)methoxy)phenyl)benzothiazole **19b.** Using the above-mentioned procedure from **10b** (70.0 mg, 0.27 mmol) 2-(bromomethyl)pyridine hydrobromide (81.9 mg, 0.32 mmol), compound **19b** was obtained as a yellow powder (38.3 mg, 40%; m.p. 190–195 °C). ^1^H NMR (300 MHz, DMSO) δ 8.61 (1H, d, *J* = 4.0 Hz), 8.12–8.01 (2H, m), 7.98–7.81 (3H, m), 7.57 (1H, d, *J* = 7.8 Hz), 7.49–7.35 (3H, m), 5.38 (2H, s OCH_2_). ^13^C NMR (151 MHz, DMSO) δ 165.91, 159.73 (d, *J_CF_* = 243.1 Hz), 155.62, 151.71 (d, *J_CF_* = 245.9 Hz), 150.29, 149.25, 148.60 (d, *J_CF_* = 10.7 Hz), 137.16, 135.71 (d, *J_CF_* = 11.7 Hz), 126.03 (d, *J_CF_* = 6.7 Hz), 124.26 (d, *J_CF_* = 3.0 Hz), 123.91 (d, *J_CF_* = 9.6 Hz), 123.27, 121.87, 115.74, 115.12 (d, *J_CF_* = 24.9 Hz), 114.37 (d, *J_CF_* = 20.0 Hz), 108.74 (d, *J_CF_* = 27.4 Hz), 71.23 (OCH_2_). Anal.calcd. for C_19_H_12_F_2_N_2_OS (Mr = 354.37): C 64.40, H 3.41, N 7.91; found: C 64.16, H 3.41, N 7.87.

6-Fluoro-2-(3-methoxy-4-(pyridin-2-yl)methoxy)phenyl)benzothiazole **19c.** Using the above-mentioned procedure from **10c** (70.0 mg, 0.25 mmol) 2-(bromomethyl)pyridine hydrobromide (75.9 mg, 0.30 mmol), compound **19c** was obtained as grey powder (62.4 mg, 66%; m.p. 152–156 °C). ^1^H NMR (300 MHz, DMSO) δ 8.60 (1H, d, *J* = 4.1 Hz), 8.12–7.97 (2H, m), 7.87 (1H, td, *J* = 7.7, 1.6 Hz), 7.67 (1H, d, *J* = 1.8 Hz), 7.56 (2H, dd, *J* = 11.0, 4.7 Hz), 7.46–7.30 (2H, m), 7.21 (1H, d, *J* = 8.5 Hz), 5.28 (2H, s, OCH_2_), 3.93 (3H, s, OCH_3_). ^13^C NMR (75 MHz, DMSO) δ 167.73 (d, *J_CF_* = 3.1 Hz), 160.11 (d, *J_CF_* = 242.6 Hz), 156.72, 150.89, 150.84, 149.86, 149.64, 137.55, 136.05 (d, *J_CF_* = 12.0 Hz), 126.35, 124.17 (d, *J_CF_* = 9.5 Hz), 123.59, 122.27, 121.27, 115.40 (d, *J_CF_* = 24.7 Hz), 113.97, 110.08, 109.11 (d, *J_CF_* = 27.3 Hz), 71.38 (OCH_2_), 56.22 (OCH_3_). Anal.calcd. for C_20_H_15_FN_2_O_2_S (Mr = 366.41): C 65.56, H 4.13, N 7.65; found: C 65.31, H 4.12, N 7.61.

2-(4-(Pyridin-2-ylmethoxy)phenyl)benzothiazole **20a.** Using the above-mentioned procedure from **11a** (120 mg, 0.53 mmol) 2-(bromomethyl)pyridine hydrobromide (160.9 mg, 0.64 mmol), compound **20a** was obtained as a beige powder (57.8 mg, 34%; m.p. 132–136 °C). ^1^H NMR (600 MHz, DMSO) δ 8.61 (1H, dd, *J* = 4.7, 1.5, 0.7 Hz), 8.11 (1H, d, *J* = 7.8 Hz), 8.05 (2H, d, *J* = 8.8 Hz), 8.01 (1H, d, *J* = 8.1 Hz), 7.86 (1H, td, *J* = 7.7, 1.7 Hz), 7.55 (1H, d, *J* = 7.8 Hz), 7.54–7.51 (1H, m), 7.45–7.40 (1H, m), 7.39–7.35 (1H, m), 7.22 (2H, d, *J* = 8.8 Hz), 5.30 (2H, s, OCH_2_). ^13^C NMR (151 MHz, DMSO) δ 166.91, 160.63, 156.17, 153.63, 149.17, 137.04, 134.24, 128.89, 126.50, 125.89, 125.12, 123.08, 122.48, 122.19, 121.76, 115.57, 70.49 (OCH_2_). Anal.calcd. for C_19_H_14_N_2_OS (Mr = 318.39): C 71.68, H 4.43, N 8.80; found: C 71.34, H 4.43, N 8.75.

2-(3-Fluoro-4-(pyridin-2-yl)methoxy)phenyl)benzothiazole **20b.** Using the above-mentioned procedure from **11b** (120 mg, 0.49 mmol) 2-(bromomethyl)pyridine hydrobromide (148.7 mg, 0.59 mmol), compound **20b** was obtained as a beige powder (36.9 mg, 22%; m.p. 136–142 °C). ^1^H NMR (300 MHz, DMSO) δ 8.61 (1H, d, *J* = 4.1 Hz), 8.14 (1H, d, *J* = 7.9 Hz), 8.04 (1H, d, *J* = 7.8 Hz), 7.96 (1H, dd, *J* = 12.0, 2.0 Hz), 7.92–7.82 (2H, m), 7.60–7.50 (2H, m), 7.42 (3H, dd, *J* = 13.0, 7.3, 4.4 Hz), 5.38 (2H, s, OCH_2_). ^13^C NMR (75 MHz, DMSO) δ 156.14, 153.83, 152.21 (d, *J_CF_* = 245.8 Hz), 149.75, 149.07 (d, *J_CF_* = 10.5 Hz), 137.66, 134.93, 130.52, 127.16, 126.74 (d, *J_CF_* = 6.6 Hz), 125.93, 124.78 (d, *J_CF_* = 3.1 Hz), 123.77, 123.18, 122.81, 122.36, 116.22, 114.92 (d, *J_CF_* = 20.1 Hz), 71.72 (OCH_2_). Anal.calcd. for C_19_H_13_FN_2_OS (Mr = 336.38): C 67.84, H 3.90, N 8.33; found: C 67.55, H 3.89, N 8.28.

2-(3-Methoxy-4-(pyridin-2-yl)methoxy)phenyl)benzothiazole **20c.** Using the above-mentioned procedure from **11c** (120 mg, 0.47 mmol) 2-(bromomethyl)pyridine hydrobromide (142.7 mg, 0.56 mmol), compound **20c** was obtained as a beige powder (99.4 mg, 61%; m.p. 110–115 °C). ^1^H NMR (400 MHz, DMSO) δ 8.61 (1H, dd, *J* = 4.8, 1.7, 0.9 Hz), 8.11 (1H, d, *J* = 7.3 Hz), 8.04 (1H, d, *J* = 7.7 Hz), 7.87 (1H, td, *J* = 7.7, 1.8 Hz), 7.70 (1H, d, *J* = 2.1 Hz), 7.60 (1H, dd, *J* = 8.4, 2.1 Hz), 7.58–7.50 (2H, m), 7.46–7.41 (1H, m), 7.38 (1H, dd, *J* = 7.5, 4.8, 1.1 Hz), 7.21 (1H, d, *J* = 8.5 Hz), 5.28 (2H, s, OCH_2_), 3.94 (3H, s, OCH_3_). ^13^C NMR (101 MHz, DMSO) δ 167.60, 156.76, 154.06, 150.82, 149.86, 149.65, 137.56, 134.81, 127.02, 126.57, 125.66, 123.59, 123.00, 122.68, 122.28, 121.30, 113.97, 110.18, 71.39 (OCH_2_), 56.23 (OCH_3_). Anal.calcd. for C_20_H_16_N_2_O_2_S (Mr = 348.42): C 68.95, H 4.63, N 8.04; found: C 68.68, H 4.62, N 8.00.

#### 3.2.3. General Procedure for Preparation of Target 1H-1,2,3-Triazole-substituted Benzothiazole Analogs **21a**–**21c**, **22a**–**22c** and **23a**–**23c**

The reaction mixture of compounds **12a**–**12c, 13a**–**13c, 14a**–**14c**, CuI (0.1 eq) and the trimethylsilyl azide (1.5 eq) was dissolved in a mixture of DMF:MeOH  =  1:1 (2 mL). The reaction mixture was stirred at 100 °C for 12 h. The solvent was removed under reduced pressure and purified by column chromatography with CH_2_Cl_2_.

6-Chloro-2-(4-((1*H*-1,2,3-triazol-4-yl)methoxy)phenyl)benzothiazole **21a.** Compound **21a** was prepared using the above-mentioned procedure from **12a** (200 mg. 0.67 mmol) and trimethylsilyl azide (132 µL, 1.00 mmol) to obtain **21a** as a beige powder (88.1 mg, 38%; m.p. 220–223 °C). ^1^H NMR (300 MHz, DMSO) δ 15.13 (1H, s, NH), 8.25 (1H, d, *J* = 2.1 Hz), 8.05–7.94 (4H, m), 7.53 (1H, dd, *J* = 8.7, 2.2 Hz), 7.21 (2H, d, *J* = 8.8 Hz), 5.29 (2H, s, OCH_2_). ^13^C NMR (151 MHz, DMSO) δ 167.96, 160.67, 152.41, 135.80, 129.48, 128.94, 126.96, 125.46, 123.58, 121.88, 115.55, 61.10 (OCH_2_). Anal.calcd. for C_16_H_11_ClN_4_OS (Mr = 342.80): C 56.06, H 3.23, N 16.34; found: C 55.84, H 3.23, N 16.26.

6-Chloro-2-(3-fluoro-4-((1*H*-1,2,3-triazol-4-yl)methoxy)phenyl)benzothiazole **21b.** Compound **21b** was prepared using the above-mentioned procedure from **12b** (200 mg, 0.63 mmol) and trimethylsilyl azide (124 µL, 0.95 mmol) to obtain **21b** as a yellow powder (35.7 mg, 15%; m.p. 205–209 °C). ^1^H NMR (400 MHz, DMSO) δ 15.10 (1H, s, NH), 8.31 (1H, d, *J* = 2.1 Hz), 8.03 (1H, d, *J* = 8.7 Hz), 7.93 (1H, dd, *J* = 11.9, 2.1 Hz), 7.91–7.86 (1H, m), 7.57 (2H, dd, *J* = 8.7, 2.2 Hz), 5.40 (2H, s, OCH_2_). ^13^C NMR (101 MHz, DMSO) δ 167.33, 152.71, 152.20 (d, *J_CF_* = 246.0 Hz), 149.08 (d, *J_CF_* = 10.2 Hz), 136.50, 130.33, 127.65, 126.35 (d, *J_CF_* = 6.7 Hz), 124.85 (d, *J_CF_* = 2.5 Hz), 124.31, 122.51, 116.28, 114.96 (d, *J_CF_* = 20.2 Hz), 62.54 (OCH_2_). Anal.calcd. for C_16_H_10_ClFN_4_OS (Mr = 360.79): C 53.27, H 2.79, N 15.53; found: C 53.08, H 2.79, N 15.46.

6-Chloro-2-(3-methoxy-4-((1*H*-1,2,3-triazol-4-yl)methoxy)phenyl)benzothiazole **21c.** Compound **21c** was prepared using the above-mentioned procedure from **12c** (200 mg, 0.61 mmol) and trimethylsilyl azide (121 µL, 0.92 mmol) to obtain **21c** as a beige powder (49.4 mg, 21%; m.p. 183–188 °C). ^1^H NMR (300 MHz, DMSO) δ 15.14 (1H, s, NH), 8.28 (1H, d, *J* = 2.1 Hz), 8.02 (2H, d, *J* = 8.7 Hz), 7.66–7.58 (2H, m), 7.55 (1H, dd, *J* = 8.7, 2.2 Hz), 7.33 (1H, d, *J* = 8.3 Hz), 5.29 (2H, s, OCH_2_), 3.88 (3H, s, OCH_3_). ^13^C NMR (75 MHz, DMSO) δ 168.63, 152.83, 150.88, 149.84, 136.36, 130.03, 127.47, 126.15, 124.10, 122.35, 121.35, 114.01, 110.14, 62.02 (OCH_2_), 56.10 (OCH_3_). Anal.calcd. for C_17_H_13_ClN_4_O_2_S (Mr = 372.83): C 54.77, H 3.51, N 15.03; found: C 54.58, H 3.51, N 14.96.

6-Fluoro-2-(4-((1*H*-1,2,3-triazol-4-yl)methoxy)phenyl)benzothiazole **22a.** Compound **22a** was prepared using the above-mentioned procedure from **13a** (200 mg, 0.71 mmol) and trimethylsilyl azide (140 µL, 1.07 mmol) to obtain **22a** as a white powder (45.7 mg, 19%; m.p. 212–215 °C). ^1^H NMR (600 MHz, DMSO) δ 15.12 (1H, s, NH), 8.08–7.98 (4H, ), 7.39 (1H, td, *J* = 9.0, 2.7 Hz), 7.23 (2H, d, *J* = 8.8 Hz), 5.31 (2H, s, OCH_2_). ^13^C NMR (75 MHz, DMSO) δ 167.62 (d, *J_CF_* = 2.8 Hz), 160.96, 160.08 (d, *J_CF_* = 242.6 Hz), 150.93, 135.96 (d, *J_CF_* = 11.9 Hz), 129.29, 126.11, 124.14 (d, *J_CF_* = 9.5 Hz), 116.04, 115.40 (d, *J_CF_* = 24.8 Hz), 109.10 (d, *J_CF_* = 27.3 Hz), 61.59 (OCH_2_). Anal.calcd. for C_16_H_11_FN_4_OS (Mr = 326.35): C 58.89, H 3.40, N 17.17; found: C 58.65, H 3.40, N 17.08.

6-Fluoro-2-(3-fluoro-4-((1*H*-1,2,3-triazol-4-yl)methoxy)phenyl)benzothiazole **22b.** Compound **22b** was prepared using the above-mentioned procedure from **13b** (200 mg, 0.66 mmol) and trimethylsilyl azide (130 µL, 0.99 mmol) to obtain **22b** as a brown powder (48.3 mg, 21%; m.p. 177–180 °C). ^1^H NMR (300 MHz, DMSO) δ 15.19 (1H, s, NH), 8.11–8.00 (2H, m), 7.96–7.84 (2H, m), 7.55 (1H, t, *J* = 8.6 Hz), 7.41 (1H, td, *J* = 9.1, 2.7 Hz), 5.40 (2H, s, OCH_2_). ^13^C NMR (75 MHz, DMSO) δ 166.42, 160.23 (d, *J_CF_* = 243.1 Hz), 150.77, 152.20 (d, *J_CF_* = 245.8 Hz), 148.82, 136.20 (d, *J_CF_* = 12.1 Hz), 126.43, 124.67 (d, *J_CF_* = 3.2 Hz), 124.41 (d, *J_CF_* = 9.6 Hz), 116.29, 115.61 (d, *J_CF_* = 24.8 Hz), 114.83 (d, *J_CF_* = 20.3 Hz), 109.22 (d, *J_CF_* = 27.5 Hz), 62.54 (OCH_2_). Anal.calcd. for C_16_H_10_F_2_N_4_OS (Mr = 344.34): C 55.81, H 2.93, N 16.27; found: C 55.59, H 2.92, N 16.19.

6-Fluoro-2-(3-methoxy-4-((1*H*-1,2,3-triazol-4-yl)methoxy)phenyl)benzothiazole **22c.** Compound **22c** was prepared using the above-mentioned procedure from **13c** (200 mg, 0.64 mmol) and trimethylsilyl azide (126 µL, 0.96 mmol) to obtain **22c** as a beige powder (50.6 mg, 22%; m.p. 196–199 °C). ^1^H NMR (300 MHz, DMSO) δ 15.10 (1H, s, NH), 8.04 (3H, dd, *J* = 9.0, 5.2 Hz), 7.63 (1H, s), 7.59 (1H, dd, *J* = 8.4, 1.8 Hz), 7.39 (1H, td, *J* = 9.1, 2.6 Hz), 7.32 (1H, d, *J* = 8.4 Hz), 5.28 (2H, s, OCH_2_), 3.87 (3H, s, OCH_3_). ^13^C NMR (75 MHz, DMSO) δ 167.23 (d, *J_CF_* = 3.1 Hz), 159.61 (d, *J_CF_* = 242.7 Hz), 150.41, 150.20, 149.36, 135.55 (d, *J_CF_* = 11.8 Hz), 128.79, 125.85, 123.66 (d, *J_CF_* = 9.3 Hz), 120.68, 114.88 (d, *J_CF_* = 24.9 Hz), 113.54, 109.55, 108.59 (d, *J_CF_* = 27.0 Hz), 61.51 (OCH_2_), 55.60 (OCH_3_). Anal.calcd. for C_17_H_13_FN_4_O_2_S (Mr = 356.37): C 57.30, H 3.68, N 15.72; found: C 57.09, H 3.67, N 15.65.

2-(4-((1*H*-1,2,3-triazol-4-yl)methoxy)phenyl)benzothiazole **23a.** Compound **23a** was prepared using the above-mentioned procedure from **14a** (200 mg, 0.75 mmol) and trimethylsilyl azide (149 µL, 1.13 mmol) to obtain **23a** as a beige powder (53.6 mg, 23%; m.p. 204–207 °C). ^1^H NMR (400 MHz, DMSO) δ 15.04 (1H, s, NH), 8.12 (1H, d, *J* = 7.7 Hz), 8.05 (2H, d, *J* = 8.8 Hz), 8.02 (1H, d, *J* = 8.4 Hz), 7.53 (1H, t, *J* = 8.2 Hz), 7.44 (1H, t, *J* = 8.1 Hz), 7.24 (2H, d, *J* = 8.8 Hz), 5.31 (2H, s, OCH_2_). ^13^C NMR (75 MHz, DMSO) δ 167.43, 160.97, 154.13, 134.73, 129.34, 127.00, 126.34, 125.62, 122.97, 122.68, 116.01, 61.65 (OCH_2_). Anal.calcd. for C_16_H_12_N_4_OS (Mr = 308.36): C 62.32, H 3.92, N 18.17; found: C 62.06, H 3.91, N 18.07

2-(3-Fluoro-4-((1*H*-1,2,3-triazol-4-yl)methoxy)phenyl)benzothiazole **23b.** Compound **23b** was prepared using the above-mentioned procedure from **14b** (200 mg, 0.71 mmol) and trimethylsilyl azide (139 µL, 1.06 mmol) to obtain **23b** as a grey powder (37.3 mg, 16%; m.p. 185–189 °C). ^1^H NMR (300 MHz, DMSO) δ 15.16 (1H, s, NH), 8.14 (1H, d, *J* = 7.3 Hz), 8.07 (1H, s), 8.04 (1H, d, *J* = 7.7 Hz), 7.98–7.85 (2H, m, *J* = 7.7, 7.1, 1.6 Hz), 7.60–7.51 (2H, m, *J* = 9.6, 6.1, 2.0 Hz), 7.46 (1H, t, *J* = 7.0 Hz), 5.40 (2H, s, OCH_2_). ^13^C NMR (75 MHz, DMSO) δ 166.27 (d, *J_CF_* = 2.7 Hz), 162.82, 153.90, 152.20 (d, *J_CF_* = 245.8 Hz), 148.83 (d, *J_CF_* = 10.6 Hz), 134.90, 127.16, 126.72 (d, *J_CF_* = 7.0 Hz), 125.93, 124.69 (d, *J_CF_* = 3.3 Hz), 123.16, 122.77, 116.26 (d, *J_CF_* = 1.6 Hz), 114.87 (d, *J_CF_* = 20.2 Hz), 62.38 (OCH_2_). Anal.calcd. for C_16_H_11_FN_4_OS (Mr = 326.35): C 58.89, H 3.40, N 17.17; found: C 58.63, H 3.39, N 17.07.

2-(3-Methoxy-4-((1*H*-1,2,3-triazol-4-yl)methoxy)phenyl)benzothiazole **23c.** Compound **23c** was prepared using the above-mentioned procedure from **14c** (200 mg, 0.68 mmol) and trimethylsilyl azide (134 µL, 1.02 mmol) to obtain **23c** as a beige powder (32.2 mg, 13%; m.p. 177–180 °C). ^1^H NMR (400 MHz, DMSO) δ 15.10 (1H, s, NH), 8.12 (1H, d, *J* = 7.7 Hz), 8.04 (1H, d, *J* = 8.0 Hz), 7.67 (1H, d, *J* = 2.0 Hz), 7.62 (1H, dd, *J* = 8.3, 2.0 Hz), 7.53 (1H, t, *J* = 7.1 Hz), 7.44 (1H, t, *J* = 7.1 Hz), 7.33 (1H, d, *J* = 8.4 Hz), 5.29 (2H, s, OCH_2_), 3.89 (3H, s, OCH_3_). ^13^C NMR (101 MHz, DMSO) δ 167.61, 154.06, 150.68, 149.85, 134.80, 127.03, 126.56, 125.67, 123.00, 122.69, 121.22, 114.03, 110.13, 62.00 (OCH_2_), 56.10 (OCH_3_). Anal.calcd. for C_17_H_14_N_4_O_2_S (Mr = 338.38): C 60.34, H 4.17, N 16.56; found: C 60.09, H 4.16, N 16.47.

#### 3.2.4. General Procedure for Preparation of Target 1-Benzyl-1,2,3-triazole-substituted Benzothiazole Analogs **24a**–**24c**, **25a**–**25c** and **26a**–**26c**

Stir a solution of benzyl chloride (1.2 eq), NaN_3_ (1.5 eq) and triethylamine (1.5 eq) in a mixture of *t*-BuOH:H_2_O = 1:1 (2 mL) at room temperature for 2 h. To reaction mixture, add corresponding propargylated compounds **12a**–**12c, 13a**–**13c, 14a**–**14c** (1 eq) and Cu(OAc)_2_ (0.2 eq). The reaction mixture was stirred at room temperature for 12 h. The solvent was removed under reduced pressure and purified by column chromatography with CH_2_Cl_2_.

6-Chloro-2-(4-((1-benzyl-*1H*-1,2,3-triazol-4-yl)methoxy)phenyl)benzothiazole **24a.** Using the above-mentioned procedure from **12a** (100 mg, 0.33 mmol) and benzyl chloride (46 µL, 0.39 mmol), compound **24a** was obtained as an orange powder (82.4 mg, 57%; m.p. 201–205 °C). ^1^H NMR (600 MHz, DMSO) δ 8.36 (1H, s), 8.31 (1H, d, *J* = 2.1 Hz), 8.06 (2H, d, *J* = 8.9 Hz), 8.03 (1H, d, *J* = 8.7 Hz), 7.58 (1H, dd, *J* = 8.6, 2.2 Hz), 7.44–7.39 (2H, m), 7.37 (3H, dd, *J* = 10.1, 4.5 Hz), 7.26 (2H, d, *J* = 8.9 Hz), 5.65 (2H, s, NCH_2_), 5.29 (2H, s, OCH_2_). ^13^C NMR (151 MHz, DMSO) δ 167.98, 160.69, 152.42, 142.51, 135.95, 135.80, 129.48, 128.93, 128.75, 128.15, 127.94, 126.97, 125.43, 124.85, 123.59, 121.89, 115.55, 61.34 (OCH_2_), 52.83 (NCH_2_). Anal.calcd. for C_23_H_17_ClN_4_OS (Mr = 432.93): C 63.81, H 3.96, N 12.94; found: C 63.60, H 3.95, N 12.89.

6-Chloro-2-(3-fluoro-4-((1-benzyl-*1H*-1,2,3-triazol-4-yl)methoxy)phenyl)benzothiazole **24b.** Using the above-mentioned procedure from **12b** (100 mg, 0.31 mmol) and benzyl chloride (43 µL, 0.37 mmol), compound **24b** was obtained as a beige powder (53.4 mg, 38%; m.p. 181–186 °C). ^1^H NMR (300 MHz, DMSO) δ 8.36 (1H, s), 8.31 (1H, d, *J* = 2.1 Hz), 8.02 (1H, d, *J* = 8.7 Hz), 7.90 (2H, t, *J* = 10.1 Hz), 7.61–7.50 (2H, m), 7.43–7.27 (5H, m), 5.63 (2H, s, NCH_2_), 5.35 (2H, s, OCH_2_). ^13^C NMR (75 MHz, DMSO) δ 167.33, 152.70, 152.18 (d, *J_CF_* = 245.9 Hz), 149.07 (d, *J_CF_* = 10.6 Hz), 142.54, 136.45 (d, *J_CF_* = 5.6 Hz), 134.63, 130.31, 129.25, 128.65, 128.44, 127.64, 126.32 (d, *J_CF_* = 7.0 Hz), 125.65, 124.83 (d, *J_CF_* = 3.1 Hz), 124.30, 122.50, 116.31 (d, *J_CF_* = 1.5 Hz), 114.94 (d, *J_CF_* = 20.2 Hz), 62.6 (OCH_2_), 53.34 (NCH_2_). Anal.calcd. for C_23_H_16_ClFN_4_OS (Mr = 450.92): C 61.26, H 3.58, N 12.43; found: C 61.07, H 3.57, N 12.37.

6-Chloro-2-(3-methoxy-4-((1-benzyl-*1H*-1,2,3-triazol-4-yl)methoxy)phenyl)benzothiazole **24c.** Using the above-mentioned procedure from **12c** (100 mg, 0.30 mmol) and benzyl chloride (41 µL, 0.36 mmol), compound **24c** was obtained as a beige powder (51.7 mg, 37%; m.p. 190–193 °C). ^1^H NMR (300 MHz, DMSO) δ 8.32 (1H, s), 8.27 (1H, d, *J* = 2.1 Hz), 8.01 (1H, d, *J* = 8.7 Hz), 7.66–7.57 (2H, m), 7.55 (1H, dd, *J* = 8.7, 2.2 Hz), 7.42–7.28 (6H, m), 5.62 (2H, s, NCH_2_), 5.23 (2H, s, OCH_2_), 3.85 (3H, s, OCH_3_). ^13^C NMR (75 MHz, DMSO) δ 168.64, 152.84, 150.88, 149.82, 142.96, 136.44, 136.36, 130.02, 129.24, 128.65, 128.47, 127.47, 126.12, 125.50, 124.10, 122.36, 121.34, 114.02, 110.06, 62.18(OCH_2_), 56.05(OCH_3_), 53.32 (NCH_2_). Anal.calcd. for C_24_H_19_ClN_4_O_2_S (Mr = 462.95): C 62.27, H 4.14, N 12.10; found: C 62.07, H 4.13, N 12.05.

6-Fluoro-2-(4-((1-benzyl-*1H*-1,2,3-triazol-4-yl)methoxy)phenyl)benzothiazole **25a.** Using the above-mentioned procedure from **13a** (100 mg, 0.35 mmol) and benzyl chloride (48 µL, 0.42 mmol), compound **25a** was obtained as a white powder (18.3 mg, 12%; m.p. 176–180 °C). ^1^H NMR (600 MHz, DMSO) δ 8.33 (1H, s), 8.06–7.99 (4H, m), 7.42–7.36 (3H, m), 7.36–7.30 (3H, m), 7.22 (2H, d, *J* = 8.8 Hz), 5.63 (2H, s, NCH_2_), 5.26 (2H, s, OCH_2_). ^13^C NMR (151 MHz, DMSO) δ 167.09 (d, *J_CF_* = 3.1 Hz), 160.51, 159.58 (d, *J_CF_* = 242.6 Hz), 150.48 (d, *J_CF_* = 1.0 Hz), 142.53, 135.94, 135.48 (d, *J_CF_* = 11.9 Hz), 128.78, 128.75, 128.15, 127.94, 125.60, 124.84, 123.65 (d, *J_CF_* = 9.5 Hz), 115.53, 114.89 (d, *J_CF_* = 24.7 Hz), 108.64 (d, *J_CF_* = 27.3 Hz), 61.32 (OCH_2_), 52.83 (NCH_2_). Anal.calcd. for C_23_H_17_FN_4_OS (Mr = 416.47): C 66.33, H 4.11, N 13.45; found: C 66.11, H 4.10, N 13.39.

6-Fluoro-2-(3-fluoro-4-((1-benzyl-*1H*-1,2,3-triazol-4-yl)methoxy)phenyl)benzothiazole **25b.** Using the above-mentioned procedure from **13b** (100 mg, 0.33 mmol) and benzyl chloride (46 µL, 0.39 mmol), compound **25b** was obtained as a white powder (69.5 mg, 48%; m.p. 205–208 °C). ^1^H NMR (300 MHz, DMSO) δ 8.34 (1H, s), 8.08–7.99 (2H, m), 7.91–7.80 (2H, m), 7.54 (1H, t, *J* = 8.5 Hz), 7.43–7.28 (6H, m), 5.61 (2H, s, NCH_2_), 5.33 (2H, s, OCH_2_). ^13^C NMR (75 MHz, DMSO) δ 160.23 (d, *J_CF_* = 243.0 Hz), 150.79 (d, *J_CF_* = 1.3 Hz), 152.20 (d, *J_CF_* = 245.8 Hz), 148.89 (d, *J_CF_* = 10.6 Hz), 142.57, 136.41, 136.21 (d, *J_CF_* = 12.0 Hz), 129.24, 128.65, 128.43, 126.55, 125.64, 124.66 (d, *J_CF_* = 3.1 Hz), 124.41 (d, *J_CF_* = 9.7 Hz), 116.33 (d, *J_CF_* = 1.7 Hz), 115.61 (d, *J_CF_* = 24.9 Hz), 114.82 (d, *J_CF_* = 20.2 Hz), 109.23 (d, *J_CF_* = 27.5 Hz), 62.68 (OCH_2_), 53.35 (NCH_2_). Anal.calcd. for C_23_H_16_F_2_N_4_OS (Mr = 434.46): C 63.58, H 3.71, N 12.90; found: C 63.39, H 3.71, N 12.84.

6-Fluoro-2-(3-methoxy-4-((1-benzyl-*1H*-1,2,3-triazol-4-yl)methoxy)phenyl)benzothiazole **25c.** Using the above-mentioned procedure from **13c** (100 mg, 0.33 mmol) and benzyl chloride (44 µL, 0.38 mmol), compound **25c** was obtained as white powder (50.4 mg, 35%; m.p. 180–183 °C). ^1^H NMR (600 MHz, DMSO) δ 8.29 (1H, s), 8.04–7.99 (2H, m), 7.60 (1H, d, *J* = 2.0 Hz), 7.56 (1H, dd, *J* = 8.4, 2.1 Hz), 7.36 (3H, m), 7.33–7.28 (4H, m), 5.60 (2H, s, NCH_2_), 5.21 (2H, s, OCH_2_), 3.83 (3H, s, OCH_3_). ^13^C NMR (75 MHz, DMSO) δ 167.75 (d, *J_CF_* = 3.3 Hz), 160.11 (d, *J_CF_* = 242.7 Hz), 150.90 (d, *J_CF_* = 1.1 Hz), 150.69, 149.83, 142.98, 136.44, 136.04 (d, *J_CF_* = 12.1 Hz), 129.24, 128.65, 128.47, 126.30, 125.49, 124.16 (d, *J_CF_* = 9.6 Hz), 121.18, 115.40 (d, *J_CF_* = 24.7 Hz), 114.04, 109.96, 109.12 (d, *J_CF_* = 27.3 Hz), 62.17 (OCH_2_), 56.04 (OCH_3_), 53.31 (NCH_2_). Anal.calcd. for C_24_H_19_FN_4_O_2_S (Mr = 446.50): C 64.56, H 4.29, N 12.55; found: C 64.37, H 4.28, N 12.50.

2-(4-((1-Benzyl-*1H*-1,2,3-triazol-4-yl)methoxy)phenyl)benzothiazole **26a.** Using the above-mentioned procedure from **14a** (100 mg, 0.38 mmol) and benzyl chloride (53 µL, 0.46 mmol), compound **26a** was obtained as a white powder (50.2 mg, 33%; m.p. 210–214 °C). ^1^H NMR (300 MHz, DMSO) δ 8.33 (1H, s), 8.11 (1H, d, *J* = 7.9 Hz), 8.03 (3H, t, *J* = 7.8 Hz), 7.57–7.48 (1H, m), 7.46–7.40 (1H, m), 7.40–7.29 (5H, m), 7.22 (2H, d, *J* = 8.9 Hz), 5.63 (2H, s, NCH_2_), 5.26 (2H, s, OCH_2_). ^13^C NMR (151 MHz, DMSO) δ 166.94, 160.49, 153.63, 142.55, 135.94, 134.22, 128.83, 128.75, 128.15, 127.94, 126.51, 125.79, 125.12, 124.84 (Tr), 122.47, 122.19, 115.50, 61.32 (OCH_2_), 52.83 (NCH_2_). Anal.calcd. for C_23_H_18_N_4_OS (Mr = 398.48): C 69.33, H 4.55, N 14.06; found: C 69.09, H 4.56, N 14.00.

2-(3-Fluoro-4-((1-benzyl-*1H*-1,2,3-triazol-4-yl)methoxy)phenyl)benzothiazole **26b.** Using the above-mentioned procedure from **14b** (100 mg, 0.35 mmol) and benzyl chloride (48 µL, 0.42 mmol), compound **26b** was obtained as white powder (40.5 mg, 27%; m.p. 198–201 °C). ^1^H NMR (300 MHz, DMSO) δ 8.36 (1H, s), 8.14 (1H, d, *J* = 7.8 Hz), 8.04 (1H, d, *J* = 8.0 Hz), 7.96–7.84 (2H, m), 7.61–7.50 (2H, m), 7.45 (1H, t, *J* = 7.3 Hz), 7.42–7.27 (5H, m), 5.64 (2H, s, NCH_2_), 5.35 (2H, s, OCH_2_). ^13^C NMR (75 MHz, DMSO) δ 165.74, 153.43, 151.70 (d, *J_CF_* = 245.8 Hz), 148.35 (d, *J_CF_* = 10.5 Hz), 142.08, 135.91, 134.43, 128.73, 127.92, 126.64, 126.22 (d, *J_CF_* = 7.0 Hz), 125.41, 125.12, 124.16, 123.92, 122.67, 122.28, 115.81, 114.36 (d, *J_CF_* = 19.6 Hz), 62.18 (OCH_2_), 52.85 (OCH_3_). Anal.calcd. for C_23_H_17_FN_4_OS (Mr = 416.47): C 66.33, H 4.11, N 13.45; found: C 66.12, H 4.10, N 13.40.

2-(3-Methoxy-4-((1-benzyl-*1H*-1,2,3-triazol-4-yl)methoxy)phenyl)benzothiazole **26c.** Using the above-mentioned procedure from **14c** (100 mg, 0.34 mmol) and benzyl chloride (47 µL, 0.41 mmol), compound **26c** was obtained as white powder (43.7 mg, 33%; m.p. 170–173 °C). ^1^H NMR (300 MHz, DMSO) δ 8.32 (1H, s), 8.12 (1H, d, *J* = 7.6 Hz), 8.03 (1H, d, *J* = 7.9 Hz), 7.66 (1H, d, *J* = 2.0 Hz), 7.61 (1H, dd, *J* = 8.3, 2.0 Hz), 7.53 (1H, t, *J* = 7.0 Hz), 7.47–7.41 (1H, m), 7.41–7.29 (5H, m), 5.63 (2H, s, NCH_2_), 5.24 (2H, s, OCH_2_), 3.87 (3H, s, OCH_3_). ^13^C NMR (151 MHz, DMSO) δ 167.11, 153.56, 150.17, 149.33, 142.51, 135.95, 134.29, 128.74, 128.15, 127.97, 126.51, 126.03, 125.15, 124.98, 122.48, 122.17, 120.70, 113.57, 109.58, 61.69 (OCH_2_), 55.56 (OCH_3_), 52.82 (NCH_2_). Anal.calcd. for C_24_H_20_N_4_O_2_S (Mr = 428.51): C 67.27, H 4.70, N 13.08; found: C 67.07, H 4.70, N 13.02.

#### 3.2.5. General Procedure for the Synthesis of Target 6-Amidino-substituted Benzothiazole Analogs **34a**–**34c**, **35a**, **35c**, **36a**–**36c**, and **37a**–**37c**

To a stirred solution of amidino-substituted 2-aminobenzenethiolate **32** or **33** (1 eq) in glacial acetic acid (3 mL), a corresponding benzaldehyde (1 eq) was added. The reaction mixture was stirred and heated under nitrogen for 3 h, then poured onto ice and made alkaline (pH 10–11) with 20% NaOH. Resulting free base was filtered, washed with water and dried. The free base was suspended in ethanol/HCl(g) (10 mL), and stirred at room temperature for 24 h. The addition of ether resulted in precipitation of products. Solid was collected by filtration, washed with anhydrous ether, and dried under vacuum.

2-(4-(2-Oxo-2-phenylethoxy)phenyl)-6-(4,5-dihydro-*1H*-imidazol-2-yl)benzothiazole hydrochloride **34a.** Compound **34a** was prepared using the above-mentioned procedure from **32** (60.0 mg, 0.28 mmol) and **28a** (67.3 mg, 0.28 mmol) to obtain **34a** as beige powder (17.5 mg, 12%; m.p. > 250 °C). ^1^H NMR (300 MHz, DMSO) δ 10.87 (2H, s, CNH), 8.86 (1H, d, *J* = 10.5 Hz), 8.23 (1H, dd, *J* = 13.2, 8.6 Hz), 8.15–7.98 (4H, m), 7.73 (1H, t, *J* = 7.4 Hz), 7.60 (2H, t, *J* = 7.5 Hz), 7.20 (2H, d, *J* = 8.9 Hz), 6.99 (1H, d, *J* = 8.7 Hz), 5.78 (2H, s, OCH_2_), 4.04 (4H, s, NCH_2_).^13^C NMR (75 MHz, DMSO) δ 194.47 (C=O), 172.26 (CNH), 165.07, 162.07, 161.84, 157.57, 135.24, 134.43, 129.90, 129.36, 128.38, 127.09, 125.58, 124.09, 123.37, 118.94, 116.14, 70.83 (OCH_2_), 44.92 (NCH_2_). Anal.calcd. for C_24_H_19_N_3_O_2_S × HCl × 1.75H_2_O (Mr = 481.48): C 59.87. H 4.92. N 8.73; found: C 59.98. H 4.83. N 8.86.

2-(3-Fluoro-4-(2-oxo-2-phenylethoxy)phenyl)-6-(4,5-dihydro-*1H*-imidazol-2-yl)benzothiazole hydrochloride **34b.** Compound **34b** was prepared using the above-mentioned procedure from **32**(60.0 mg, 0.28 mmol) and **28b** (72.3 mg, 0.28 mmol) to obtain **34b** as brown powder (19.8 mg, 14%; m.p. >240 °C). ^1^H NMR (400 MHz, DMSO) δ 11.30 (2H, s, CNH), 9.12 (1H, s), 8.32 (1H, d, *J* = 9.9 Hz), 8.23 (1H, d, *J* = 8.6 Hz), 8.09–8.00 (3H, m), 7.95–7.87 (1H, m), 7.73 (1H, t, *J* = 7.4 Hz), 7.61 (2H, t, *J* = 7.7 Hz), 7.37 (1H, t, *J* = 8.7 Hz), 5.91 (2H, OCH_2_), 4.03 (4H, s, NCH_2_). ^13^C NMR (151 MHz, DMSO) δ 193.56 (C=O), 170.49 (CNH), 164.11, 156.73, 151.48 (d, *J_CF_* = 245.9 Hz), 149.30 (d, *J_CF_* = 10.7 Hz), 134.80, 134.04, 133.99, 128.86, 127.90, 126.95, 125.38 (d, *J_CF_* = 6.4 Hz), 124.74 (d, *J_CF_* = 2.3 Hz), 124.04, 122.93, 118.77, 115.74, 114.93 (d, *J_CF_* = 20.2 Hz), 70.92 (OCH_2_), 44.31 (NCH_2_). Anal.calcd. for C_24_H_18_FN_3_O_2_S × HCl × 1.5H_2_O (Mr = 494.97): C 58.24. H 4.48. N 8.49; found: C 58.32. H 4.56. N 8.37.

2-(3-Methoxy-4-(2-oxo-2-phenylethoxy)phenyl)-6-(4,5-dihydro-*1H*-imidazol-2-yl)benzothiazole hydrochloride **34c.** Compound **34c** was prepared using the above-mentioned procedure from **32** (60.0 mg, 0.28 mmol) and **28c** (75.7 mg, 0.28 mmol) to obtain **34c** as beige powder (40.5 mg, 26%; m.p. >240 °C). ^1^H NMR (400 MHz, DMSO) δ 10.74 (2H, s, CNH), 8.81 (1H, s), 8.28 (1H, d, *J* = 8.6 Hz,), 8.11–8.03 (3H, m), 7.76–7.70 (2H, m), 7.66 (1H, dd, *J* = 8.4, 2.1 Hz), 7.60 (2H, t, *J* = 7.7 Hz), 7.11 (1H, d, *J* = 8.6 Hz), 5.78 (2H, s, OCH_2_), 4.06 (4H, s, NCH_2_), 3.96 (3H, s, OCH_3_). ^13^C NMR (101 MHz, DMSO) δ 194.44 (C=O), 172.42 (CNH), 170.96, 165.16, 157.53, 151.59, 149.71, 135.34, 134.71, 134.41, 129.35, 128.39, 127.04, 125.69, 123.99, 123.41, 121.91, 118.96, 113.96, 110.68, 71.08 (OCH_2_), 56.32 (OCH_3_), 44.96 (NCH_2_). Anal.calcd. for C_25_H_21_N_3_O_3_S × HCl × 3.5H_2_O (Mr = 538.53): C 55.76. H 5.33. N 7.80; found: C 55.83. H 5.26. N 7.69.

2-(4-(Pyridin-2-ylmethoxy)phenyl)-6-(4,5-dihydro-*1H*-imidazol-2-yl)benzothiazole hydrochloride **35a.** Compound **35a** was prepared using the above-mentioned procedure from **32** (60.0 mg, 0.28 mmol) and **29a** (59.7 mg, 0.28 mmol) to obtain **35a** as yellow powder (19.2 mg, 15%; m.p. > 250 °C). ^1^H NMR (600 MHz, DMSO) δ 10.99 (2H, s, NCH), 8.94 (1H, d, *J* = 1.6 Hz), 8.84 (1H, d, *J* = 4.8 Hz), 8.35 (1H, t, *J* = 7.6 Hz), 8.24 (1H, d, *J* = 8.6 Hz), 8.17 (3H, dd, *J* = 13.6, 5.2 Hz), 7.96 (1H, d, *J* = 7.8 Hz), 7.83–7.78 (1H, m), 7.31 (2H, d, *J* = 8.8 Hz), 5.55 (2H, s, OCH_2_), 4.04 (4H, s, NCH_2_). ^13^C NMR (151 MHz, DMSO) δ 171.56 (NCH), 164.44, 160.82, 157.00, 152.92, 145.14, 142.49, 134.73, 129.58, 126.68, 125.63, 125.09, 124.12, 123.72, 122.89, 118.53, 115.82, 67.63 (OCH_2_), 44.39 (NCH_2_). Anal.calcd. for C_22_H_18_N_4_OS × HCl × 1.25H_2_O (Mr = 445.45): C 59.32. H 4.86. N 12.58; found: C 58.58. H 4.72. N 12.73.

2-(3-Methoxy-4-(pyridin-2-ylmethoxy)phenyl)-6-(4,5-dihydro-*1H*-imidazol-2-yl)benzothiazole hydrochloride **35c.** Compound **35c** was prepared using the above-mentioned procedure from **32** (60.0 mg, 0.28 mmol) and **29c** (68.1 mg, 0.28 mmol) to obtain **35c** as yellow powder (24.1 mg, 19%; m.p. 219–223 °C). ^1^H NMR (300 MHz, DMSO) δ 10.93 (2H, s, CNH), 8.91 (1H, d, *J* = 1.6 Hz), 8.74 (1H, d, *J* = 4.4 Hz), 8.27 (1H, d, *J* = 8.6 Hz), 8.15 (2H, d, *J* = 8.8 Hz), 7.82–7.68 (3H, m), 7.68–7.57 (1H, m), 7.29 (1H, d, *J* = 8.3 Hz), 5.43 (2H, s, OCH_2_), 4.05 (4H, s, NCH_2_), 3.95 (3H, s, OCH_3_). ^13^C NMR (75 MHz, DMSO) δ 172.27 (CNH), 165.00, 157.45, 154.68, 151.29, 149.96, 147.20, 140.96, 135.31, 127.14, 126.19, 124.87, 124.15, 123.80, 123.40, 122.07, 119.05, 114.34, 110.60, 69.75 (OCH_2_), 56.33 (OCH_3_), 44.91 (NCH_2_). Anal.calcd. for C_23_H_20_N_4_O_2_S × HCl × H_2_O (Mr = 470.97): C 58.65, H 4.92, N 11.90; found: C 58.43, H 4.80, N 12.09.

2-(4-((1*H*-1,2,3-triazol-4-yl)methoxy)phenyl)-6-(4,5-dihydro-*1H*-imidazol-2-yl)benzothiazole hydrochloride **36a.** Compound **36a** was prepared using the above-mentioned procedure from **32** (60.0 mg, 0.28 mmol) and **30a** (56.9 mg, 0.28 mmol) to obtain **36a** as an orange powder (36.5 mg, 29%; m.p. 218–221 °C). ^1^H NMR (600 MHz, DMSO) δ 11.00 (2H, s, CNH), 8.94 (1H, d, *J* = 1.7 Hz), 8.23 (1H, d, *J* = 8.6 Hz), 8.18 (1H, dd, *J* = 8.7, 1.8 Hz), 8.12 (2H, d, *J* = 8.9 Hz), 8.04 (1H, s), 7.30–7.25 (2H, d, *J* = 8.9 Hz), 5.33 (s, 3H), 4.04 (s, 4H). ^13^C NMR (151 MHz, DMSO) δ 171.70 (CNH), 164.43, 161.27, 157.04, 134.69, 129.47, 126.67, 125.09, 123.69, 122.81, 118.45, 115.70, 61.15 (OCH_2_), 44.38 (NCH_2_). Anal.calcd. for C_19_H_16_N_6_OS × HCl × 1.5H_2_O (Mr = 439.92): C 51.87. H 4.58. N 19.10; found: C 51.68. H 4.65. N 18.98.

2-(3-Fluoro-4-((1*H*-1,2,3-triazol-4-yl)methoxy)phenyl)-6-(4,5-dihydro-*1H*-imidazol-2-yl)benzothiazole hydrochloride **36b**. Compound **36b** was prepared using the above-mentioned procedure from **32** (60.0 mg, 0.28 mmol) and **30b** (61.9 mg, 0.28 mmol) to obtain **36b** as a brown powder (22.5 mg, 17%; m.p. 234–237 °C). ^1^H NMR (600 MHz, DMSO) δ 10.98 (1H, s, CNH), 10.96 (1H, s, CNH), 8.94 (1H, d, *J* = 5.7 Hz), 8.25 (1H, d, *J* = 8.6 Hz), 8.19–8.13 (1H, m), 8.08 (1H, s), 8.00 (1H, dd, *J* = 11.7, 2.1 Hz), 7.98 (1H, dd, *J* = 8.6, 1.9 Hz), 7.60 (1H, t, *J* = 8.6 Hz), 5.42 (2H, s, OCH_2_), 4.04 (4H, s, NCH_2_). ^13^C NMR (101 MHz, DMSO) δ 171.05 (CNH), 164.94, 157.28, 152.18 (d, *J_CF_* = 246.2 Hz), 149.69 (d, *J_CF_* = 10.7 Hz), 135.40, 127.24, 125.91 (d, *J_CF_* = 6.7 Hz), 125.45 (d, *J_CF_* = 2.5 Hz), 124.31, 123.57, 119.26, 116.32, 115.35 (d, *J_CF_* = 20.2 Hz), 62.46 (OCH_2_), 44.93 (NCH_2_). Anal.calcd. for C_19_H_15_FN_6_OS × HCl × 1.5H_2_O (Mr = 457.91): C 49.84. H 4.18. N 18.35; found: C 49.95. H 4.10. N 18.23.

2-(3-Methoxy-4-((1H-1,2,3-triazol-4-yl)methoxy)phenyl)-6-(4,5-dihydro-*1H*-imidazol-2-yl)benzothiazole hydrochloride **36c**. Compound **36c** was prepared using the above-mentioned procedure from **32** (60.0 mg, 0.28 mmol) and **30c** (65.3 mg, 0.28 mmol) to obtain **36c** as beige powder (12.7 mg, 9%; m.p. >240 °C). ^1^H NMR (600 MHz, DMSO) δ 10.66 (2H, s, CNH), 8.77 (1H, s), 8.28 (1H, d, *J* = 8.6 Hz), 8.05 (1H, dd, *J* = 8.6, 1.4 Hz), 7.91 (1H, s), 7.73 (1H, dd, *J* = 8.4, 2.1 Hz), 7.70 (1H, d, *J* = 1.6 Hz), 7.38 (1H, t, *J* = 13.0 Hz), 5.31 (2H, s, OCH_2_), 4.05 (4H, s, NCH_2_), 3.89 (3H, s, OCH_3_). ^13^C NMR (151 MHz, DMSO) δ 171.37 (CNH), 164.19, 156.44, 148.80, 135.28, 134.29, 125.90, 124.35, 123.75, 122.84, 122.37, 120.97, 117.90, 112.93, 109.31, 61.00 (OCH_2_), 55.09 (OCH_3_), 43.91 (NCH_2_). Anal.calcd. for C_19_H_18_N_6_O_2_S × HCl × H_2_O (Mr = 460.94): C 52.11. H 4.59. N 18.23; found: C 51.99. H 4.67. N 18.13.

2-(4-((1-Benzyl-*1H*-1,2,3-triazol-4-yl)methoxy)phenyl)-6-(4,5-dihydro-*1H*-imidazol-2-yl)benzothiazole hydrochloride **37a**. Compound **37a** was prepared using the above-mentioned procedure from **32** (60.0 mg, 0.28 mmol) and **31a** (82.1 mg, 0.28 mmol) to obtain **37a** as a yellow powder (32.5 mg, 21%; m.p. 153–156 °C). ^1^H NMR (600 MHz, DMSO) δ 10.89 (2H, s, NCH), 8.89 (1H, d, *J* = 1.6 Hz), 8.35 (1H, s), 8.24 (1H, d, *J* = 8.6 Hz), 8.15–8.10 (3H, m), 7.41–7.36 (2H, m), 7.34 (3H, dd, *J* = 7.1, 5.0 Hz), 7.26 (2H, d, *J* = 8.9 Hz), 5.63 (2H, s, NCH_2_), 5.28 (2H, s, OCH_2_), 4.04 (4H, s, NCH_2_). ^13^C NMR (151 MHz, DMSO) δ 171.74 (CNH), 164.55, 161.30, 157.07, 142.42, 135.94, 134.72, 129.46, 128.75, 128.15, 127.95, 126.60, 125.06, 124.90, 123.61, 122.86, 118.45, 115.69, 61.40 (OCH_2_), 52.84 (NCH_2_), 44.42 (NCH_2_). Anal.calcd. for C_26_H_22_N_6_OS × HCl × 2.25H_2_O (Mr = 543.55): C 57.45. H 5.10. N 15.46; found: C 57.32. H 5.19. N 15.32.

2-(3-Fluoro-4-((1-benzyl-*1H*-1,2,3-triazol-4-yl)methoxy)phenyl)-6-(4,5-dihydro-*1H*-imidazol-2-yl)benzothiazole hydrochloride **37b**. Compound **37b** was prepared using the above-mentioned procedure from **32** (60.0 mg, 0.28 mmol) and **31b** (87.2 mg, 0.28 mmol) to obtain **37b** as a yellow powder (44.9 mg, 28%; m.p. >240 °C). ^1^H NMR (400 MHz, DMSO) δ 10.97 (2H, s, CNH), 8.93 (1H, d, *J* = 1.6 Hz), 8.39 (1H, s), 8.25 (1H, d, *J* = 8.6 Hz), 8.16 (1H, dd, *J* = 8.7, 1.8 Hz), 8.01–7.94 (2H, m), 7.60 (1H, t, *J* = 8.8 Hz), 7.43–7.27 (6H, m), 5.64 (2H, s, NCH_2_), 5.37 (2H, s, OCH_2_), 4.04 (4H, s, NCH_2_). ^13^C NMR (75 MHz, DMSO) δ 171.07 (CNH), 165.04, 157.29, 152.24 (d, *J_CF_* = 245.1 Hz), 149.70 (d, *J_CF_* = 10.7 Hz), 142.46, 136.40, 135.42, 129.25, 128.66, 128.45, 127.17, 125.70, 125.44, 124.23, 123.59, 119.25, 116.37 (d, *J* = 1.3 Hz), 115.34 (d, *J_CF_* = 20.4 Hz), 62.71 (OCH_2_), 53.35 (NCH_2_), 44.94 (NCH_2_). Anal.calcd. for C_26_H_21_FN_6_OS × HCl × 2H_2_O (Mr = 557.04): C 56.06. H 4.70. N 15.09; found: C 55.93. H 4.79. N 15.00.

2-(3-Methoxy-4-((1-benzyl-*1H*-1,2,3-triazol-4-yl)methoxy)phenyl)-6-(4,5-dihydro-*1H*-imidazol-2-yl)benzothiazole hydrochloride **37c**. Compound **37c** was prepared using the above-mentioned procedure from **32** (60.0 mg, 0.28 mmol) and **31c** (90.5 mg, 0.28 mmol) to obtain **37c** as a yellow powder (32.5 mg, 19%; m.p. >240 °C). ^1^H NMR (400 MHz, DMSO) δ 10.74 (2H, s, CNH), 8.81 (1H, d, *J* = 1.7 Hz), 8.34 (1H, s), 8.28 (1H, d, *J* = 8.6 Hz), 8.08 (1H, dd, *J* = 8.6, 1.8 Hz), 7.73 (1H, dd, *J* = 8.4, 2.1 Hz), 7.69 (1H, d, *J* = 2.0 Hz), 7.43–7.29 (6H, m), 5.64 (2H, s, NCH_2_), 5.27 (2H, s, OCH_2_), 4.06 (4H, s, NCH_2_), 3.88 (3H, s, OCH_3_). ^13^C NMR (101 MHz. DMSO) δ 172.43 (CNH), 165.19, 151.55, 142.86, 136.43, 135.34, 129.26, 128.49, 127.03, 125.56, 123.97, 123.42, 122.04, 118.97, 114.04, 110.33, 62.20 (OCH_2_), 56.12 (OCH_3_), 53.33 (NCH_2_), 44.96 (NCH_2_). Anal.calcd. for C_27_H_24_N_6_O_2_S × HCl × 3H_2_O (Mr = 587.09): C 55.24, H 5.32, N 14.31; found: C 55.36, H 5.38, N 4.17.

### 3.3. Antiproliferative Activity In Vitro

The growth inhibition activity was assessed according to the slightly modified procedure performed at the National Cancer Institute, Developmental Therapeutics Program [56].

#### 3.3.1. Cell Lines

Examined compounds were dissolved in DMSO (1 × 10^−2^ M). The experiments were carried out on seven human tumor cell lines and two normal cell lines. The following cell lines were used: HeLa (human cervical adenocarcinoma; purchased from ATCC), CaCo-2 (human colorectal adenocarcinoma), HuT78 (T-cell lymphoma), THP-1 (acute monocytic leukemia), SW620 (colorectal adenocarcinoma, metastatic), MDA-MB-231 (human breast adenocarcinoma), HL60 (promyelocytic leukemia cell line), foreskin fibroblast cells (BJ) and MDCK1 (Madine–Darby canine kidney fibroblast like cells). MDCK1 cells were used between 24 and 26 passages.

#### 3.3.2. Cell Culturing

Adherent cells were cultured in the Dulbecco’s modified Eagle medium—DMEM (Gibco, EU) supplemented with 10 % heat-inactivated fetal bovine serum (FBS, Gibco, EU), 2 mM glutamine, and 100 U/0.1 mg penicillin/streptomycin. Cells on suspension were cultured in RPMI 1640 (Gibco, EU) medium supplemented with 10 % FBS (Gibco, EU), 2 mM glutamine, 1 mM sodium pyruvate, 10 mM HEPES. Cells were grown in humidified atmosphere under the conditions of 37 °C/5% of CO_2_ gas in the CO_2_ incubator (IGO 150 CELLlife^TM^, JOUAN, Thermo Fisher Scientific**,** Waltham, MA, USA). A erythrosin B (Sigma-Aldrich, St. Louis, MO, USA) dye exclusion method was used to assess cell viability before plating.

#### 3.3.3. Proliferation Assay

Adherent cells (HeLa, CaCo-2, MCF-7 and MDCK-1) were plated in 96-well flat bottom plates (Greiner, Frickenhausen, Austria) at a concentration of 2 × 10^4^ cells/mL. Suspension cells (THP-1 and HuT78) were plated in 96-well microtiter plates (Sarstead, Newton, USA) at a concentration of 1 × 10^5^ cells/mL. Twenty-four hours later, cells were treated with test agents in five 10-fold dilutions (10^−7^ to 10^−4^ M) and incubated for further 72 h. Working dilutions were freshly prepared on the day of testing. The solvent was also tested for eventual inhibitory activity by adjusting its concentration to be the same as in working concentrations. After 72 h of incubation, the cell growth rate was evaluated by performing the MTT assay, which detects dehydrogenase activity in viable cells [57]. For this purpose, upon completion of the incubation period, growth medium was discarded and 50 μL of MTT was added to each well at a concentration of 5 mg/mL. After four hours of incubation at 37 °C, water insoluble MTT-formazan crystals were dissolved in 150 μL of dimethyl-sulfoxide (DMSO) for adherent cells, and in 10 % SDS with 0.01 M/L HCl for cells grown in suspension. The absorbance (OD, optical density) was measured on a microplate reader (iMark, BIO RAD, Hercules, CA, USA) at 595 nm.

Percent of life cells was calculated as follows: % = OD (sample)–OD (background)/OD (control)–OD (background) × 100.

Optical density (OD) of background for adherent cells is the OD of MTT solution and DMSO; OD (background) for suspension cells is OD of the culture medium with MTT and 10% SDS with 0.01 M/L HCl; OD (control) is the OD of the cells growth without tested compounds.

The results were expressed as GI_50_, a concentration necessary for 50% of inhibition. Calculation of GI_50_ value curves and QC analysis is performed by using the Excel tools and GraphPadPrism software (La Jolla, CA), v. 5.03. Briefly, individual concentration effect curves are generated by plotting the logarithm of the concentration of tested compounds(X) vs. corresponding percent inhibition values (Y) using least squares fit. The best fit GI_50_ values are calculated using Log (inhibitor) versus normalized response—Variable slope equation, where Y Ľ 100/(1 ţ 10 ((LogIC_50_ _ X) * HillSlope)). QC criteria parameters (Z0, S:B, R2, HillSlope) were checked for every GI_50_ curve.

#### 3.3.4. Cell Cycle Analysis

The HuT78 cells were plated in 6-well plates at a concentration of 5 × 10^5^ cells per well and treated 24 h and 48 h with selected compounds **36c, 42a, 42c, 45a, 45b, 45c** and **46c** at a concentration of 5 μM. After drug treatment, the cells were fixed with ice-cold 70% ethanol in phosphate-buffered saline (PBS) and incubated with 0.3 μg/mL propidium iodide for 30 min at room temperature. Before being analyzed by flow cytometry (BD FACSCalibur, Becton Dickinson, San Jose, CA, SAD), samples were treated with 0.4 μg/mL RNase A for 5 min at room temperature. The resultant DNA histograms were generated and analyzed using FlowJo 7.6 software (Treestar, Inc, Ashland, OR, USA). Experiments were done in duplicate and the quantitative data are reported as average value ± standard deviation. Comparisons between control (non-treated) and treated groups were done using one-way analysis of variance (ANOVA) with Tukey–Kramer’s post hoc test with MedCalc statistical program. *P*-value less than 0.05 was considered statistically significant.

#### 3.3.5. Measurement of Mitochondrial Membrane Potential (∆Ψm)

Changes in the (∆Ψm) were measured using TMRE (Tetramethylrhodamine, Ethyl Ester, Perchlorate) dye. In brief, tested cells (HuT78) were plated in 6-well plates at a concentration of 5 × 10^5^ cells per well and treated with 5 μM of compounds **36c, 42a, 42c, 45a, 45b, 45c**, and **46c**. After 48 h of treatment, cells were collected, centrifuged 6 min at 1100 rpm, and stained with 200 nM TMRE dye according to the kit protocol (TMRE Mitochondrial Membrane Potential Assay Kit, abcam, Cambridge, UK). Positive control cells were treated with 20 μM FCCP (carbonyl cyanide-p-trifluoromethoxyphenylhydrazone) for 10 min. Cells were analyzed by flow cytometry (BD FACSCalibur, Becton Dickinson, San Jose, CA, SAD) and FlowJo software (FlowJo, LLC, Ashland, OR, USA).

#### 3.3.6. Determination of Apoptosis

Proapoptotic potential of compounds was tested on HuT78 cells using Alexa Fluor 488 annexin V and propidium iodide (Alexa Fluor 488 annexin V/Dead Cell Apoptosis Kit, Invitrogen, Thermo Fisher Scientific, Inc., Waltham, MA, USA). Cells were plated in 6-well plates at a concentration 5 × 10^5^ cells/well and treated for 24 and 48 h with 5 µM **36c**, **42c**, **45a**, **45b**, **45c**, and **46c**. After incubation, cells were collected and centrifuged at 1100 rpm for 6 min, stained according to the manufacturer’s protocol and analyzed by flow cytometry (BD FACSCalibur, Becton Dickinson, San Jose, CA, USA) using FlowJo software (FlowJo, LLC, Ashland, OR, USA).

### 3.4. QSAR

QSAR analysis was performed on the anticancer activity against the MDCK-1 cell and Hut-78 cell line. Anticancer activities were converted in the form of the logarithm (logIC_50_). For the inactive compounds, whose IC_50_ values were estimated as 100, logIC_50_ was set to 2.

The 3D structures were optimized using molecular mechanics force fields (MM+) [58] using the HyperChem 8.0 (HyperCube, Inc., Gainesville, FL, USA). Subsequently, all structures were submitted to geometry optimization using the semi-empirical AM1 method [59]. The 2D and 3D molecular descriptors used in this study were calculated using ADMEWORKS ModelBuilder 7.9.1.0 (Fujitsu Kyushu Systems Limited, Fukuoka, Japan). Employing the QSARINS-Chem 2.2.1 (University of Insubria, Varese, Italy) [60], descriptors with a constant value for more than 80%, and descriptors that were too inter-correlated (>70%) were excluded. The final number of descriptors selected for the generation of models was 455. Generation of QSAR models was obtained by the Genetic Algorithm (GA) using QSARINS. The models were assessed by fitting criteria; internal cross-validation using the leave-one out (LOO) method; and external validation. The robustness of QSAR models was tested by the Y-randomisation test. Investigation of the applicability domain of the prediction model was performed by Williams plots (plotting residuals vs. leverage of training compounds) in order to identify the outliers and influential chemicals. The predicted data for chemicals with leverage values higher than the warning leverage (*h**) must be considered with caution. The warning leverage *h** is defined as 3*p*′/*n*, where *n* is the number of training compounds and *p*′ is the number of model parameters [43].

## 4. Conclusions

6-Halogen-substituted and 6-unsubstituted benzothiazoles were prepared by condensation of corresponding 4-hydroxybenzaldehydes and 2-aminotiophenoles and subsequent *O*-alkylation with halides to synthesize benzothiazoles **15a**–**20a**, **15b**–**20b**, and **15c**–**20c** linked via phenoxymethylene to the aromatic units. 1,2,3-Triazole-substituted benzothiazoles **21a**–**26a**, **21b**–**26b** and **21c**–**26c** were prepared by regioselective copper(I) catalyzed cycloaddition from corresponding propargylated benzothiazole intermediates and azides. 6-Imidazolyl benzothiazoles **34a**–**34c, 35a, 35c**, **36a**–**36c**, **37a**–**37c** and 6-pyrimidinyl benzothiazoles **38a**, **38b**, **39c**, **40a**–**40c**, and **41a**, **41c** were prepared by cyclocondensation of 5-amidino-2-aminothiophenoles and corresponding benzaldehydes.

We found that the antiproliferative capacity of the tested compounds varied (after 72 h of exposure, IC_50_ ranged from 1.4 × 10^−6^ M to >100 × 10^−6^ M). The majority of compounds from the non-substituted and halogen-substituted benzothiazole series did not exhibit antiproliferative activity on tested tumor cell lines. From the amidine series, 6-imidazolyl benzothiazole analogs showed strong antiproliferative activity on tested tumor cell lines; however, they were also toxic on normal cells, except for **36c**. The introduction of the 1*H*-1,2,3-triazole substituent in the benzothiazoles **36a**–**36c** resulted in reduced cytotoxicity against both MDCK1 and BJ control cell lines, while maintaining excellent growth-inhibitory effect on HuT78 cells with IC_50_ values of 4.4 µM for **36a**, 1.8 µM for **36b** and 1.6 µM for **36c** and selectivity index (SI) of 9, 18 and 94, respectively. Among benzimidazole amidines, **45a** (IC_50_ = 4.8 µM), **45b** (IC_50_ = 5.5 µM), and **45c** (IC_50_ = 4.1 µM) with 1-benzyl-1,2,3-triazole substituent, as well as **46c** (IC_50_ = 5.1 µM) containing morpholinoethyl-1,2,3-triazole, demonstrated the strongest antiproliferative activity, with SI of 12, 15, 24 and 20, respectively.

The predictive quantitative structure–activity relationship (QSAR) models have been obtained for cytotoxic effects on non-tumor MDCK-1 cells and T-cell lymphoma (HuT78) cells. QSAR analysis showed that the stronger inhibition against MDCK-1 cells depended on larger substituents, the higher 3D distribution of atomic mass at the 6-position of benzothiazoles, the presence of a sulphur atom in the benzothiazole instead of a nitrogen atom in the benzimidazole, and a sulphur atom at topological distance 4 from the fluorine atom of benzothiazoles. The presence of atoms with higher atomic mass and polarizability, such as the sulphur atom and the absence of atoms with higher van der Waals volume at topological distances 8 from the atom at position 6 of the benzothiazole, implied greater activity against HuT78.

Cell cycle perturbation assays on the HuT78 cells treated with **36c**, **42c**, **45a**–**45c**, and **46c** showed accumulation of cells in the G_2_/M and subG0/G1 phase compared to non-treated cells. Annexin-V binding flow cytometry evaluations showed that 48 h post-treatment with **36c**, **42c**, **45a**–**45c**, and **46c** the number of apoptotic cells increased. Flow cytometric analysis showed changes in mitochondrial membrane potential, suggesting that the disruption of mitochondrial membrane potential produced by **36c**, **42c**, **45a**–**45c**, and **46c** can lead to cytotoxicity and cell death by apoptosis and/or necrosis.

## Data Availability

Not applicable.

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
