# Peer review of "Synthesis, Antiproliferative Evaluation and QSAR Analysis of Novel Halogen- and Amidino-Substituted Benzothiazoles and Benzimidazoles"

_ijms, 2022, doi:10.3390/ijms232415843_

Round 1
Reviewer 1 Report
The review is very goog
Author Response
1. The manuscript should undergo extensive English revisions
The manuscript was reviewed by a native English-speaking colleague and the English language is now approved as suggested by the reviewer.
Reviewer 2 Report
This manuscript is generally well structured, and I think it enriches the current research. Nevertheless, it does require some modifications, which I have outlined in the comments below:
· The authors should compare the growth-inhibitory effects of the compounds with that of an FDA-approved anticancer drug on the same cell lines
· Selectivity indices for the compounds - compared to both normal cell lines (MDCK-1 and BJ) - should be provided in additional tables
· The cytotoxicity of all 77 compounds was tested against the cancer cell lines HeLa, CaCo-2, HuT78. What was the reason for choosing cell lines such as: SW620, MDA-MB-231, HL-60, THP1 to test the 12 most active compounds?
· Line 109: there is 16ab-16c; should be: 16a-16c
· Lines 214-215, 548, 621-622, 1083: Compound numbers should be in bold
· Lines 268, 277, 310, 1270, 1284: there is: 5 µmol dm-3; should be: 5 µM
· Lines 1220, 1239: there is: Hut78; should be: HuT78
· Lines 1221, 1239: Why the authors mention the cell lines: Burkitt lymphoma (Raji) and CCRF-CEM
· Lines 1248: there is: mg/ml; should be: mg mL-1
· Line 1250: there is: mol/L; should be: M L-1
· Lines 1271, 1274: there is: µg/ml; should be: µg mL-1
Author Response
This manuscript is generally well structured, and I think it enriches the current research. Nevertheless, it does require some modifications, which I have outlined in the comments below:
1. The authors should compare the growth-inhibitory effects of the compounds with that of an FDA-approved anticancer drug on the same cell lines.
The reviewer is right at this point. 5-Fluorouracil (5-FU), as a well-known chemotherapeutic agent, was tested under the same experimental conditions as newly synthesized compounds. Obtained IC50 values are included in Tables 1 and 2. Additionally, the inhibitory effect of novel halogen- and amidino-substituted benzothiazoles and benzimidazoles was compared with that of 5-FU in Results and Discussion section (p. 6, lines 181, 189-190, p. 7, lines 208-210).
2. Selectivity indices for the compounds - compared to both normal cell lines (MDCK-1 and BJ) - should be provided in additional tables.
The calculated values of the selectivity index for both MDCK-1 and BJ normal cell lines are presented in Tables S1-S3 (Supplementary Material) as suggested by the reviewer.
3. The cytotoxicity of all 77 compounds was tested against the cancer cell lines HeLa, CaCo-2, HuT78. What was the reason for choosing cell lines such as: SW620, MDA-MB-231, HL-60, THP1 to test the 12 most active compounds?
Given the observed significantly more pronounced effects on tumor cells of hematopoietic origin and tumor cells of lymphoid origin compared to the effects on tumor cells of epithelial origin, we decided to test the selectivity of the effects of 12 compounds with the best inhibitory effects on an expanded panel of tumor cell lines. Testing of selected compounds on this panel of cells confirmed that tumor cells of epithelial origin are more resistant to the toxic effects of the newly synthesized compounds compared to leukemia and lymphoma cell lines.
4. Line 109: there is 16ab-16c; should be: 16a-16c
It is now corrected.
5. Lines 214-215, 548, 621-622, 1083: Compound numbers should be in bold
It is now corrected
6. Lines 268, 277, 310, 1270, 1284: there is: 5 µmol dm-3; should be: 5 µM
It is now corrected.
7. Lines 1220, 1239: there is: Hut78; should be: HuT78
It is now corrected.
8. Lines 1221, 1239: Why the authors mention the cell lines: Burkitt lymphoma (Raji) and CCRF CEM
An error occurred in specifying the cell lines on which the tests were performed. This is now corrected.
9. Lines 1248: there is: mg/ml; should be: mg mL-1
It is now corrected.
10. Line 1250: there is: mol/L; should be: M L-1
It is now corrected.
11. Lines 1271, 1274: there is: µg/ml; should be: µg mL-1
It is now corrected.
Reviewer 3 Report
The paper titled “Synthesis, Antiproliferative Evaluation and QSAR Analysis of 2 Novel Halogen- and Amidino-Substituted Benzothiazoles and 3 Benzimidazoles” demonstrates synthesis and pretty high activity in the raw of some benzothiazole derivatives. While the synthetic part seems to be quite reliable, reproducible, but fairly standard, the results of biological tests seem to be very successful, demonstrating levels of inhibition against cancer cells at a level close to 1 micromol. Another important feature that is often overlooked in research is high selectivity, in other words, low activity against normal cells. I think that the article is well written, fits the scope of the journal and can be published after a few corrections.
1) While all other spectra look pretty good, 1H NMR spectra of compounds 19b, 20b, 24a, 24b, 26a contain some unpurities.
2) On the line 531 it was mentioned that TMS is used as internal standard, but on the next line written that chemical shifts corrected on the basis of residual solvent peaks. I think that one of these thesises should be removed as contradictory. Generally, if TMS added it uses as internal standard to correct chemical shifts.
3) On the scheme 4 in the conditions (i) should form not the benzo[d]thiazole but its dihydro derivative: 2,3-dihydrobenzo[d]thiazole. How do you explain this? Is there is an oxidation with oxygen? Do you observe the intermediate dihydro product? Or some oxidant was lost in procedure? On the scheme 1 for this purpose Na2S2O5 was used (conditions iii).
Author Response
The paper titled “Synthesis, Antiproliferative Evaluation and QSAR Analysis of 2 Novel Halogen- and Amidino-Substituted Benzothiazoles and 3 Benzimidazoles” demonstrates synthesis and pretty high activity in the raw of some benzothiazole derivatives. While the synthetic part seems to be quite reliable, reproducible, but fairly standard, the results of biological tests seem to be very successful, demonstrating levels of inhibition against cancer cells at a level close to 1 micromol. Another important feature that is often overlooked in research is high selectivity, in other words, low activity against normal cells. I think that the article is well written, fits the scope of the journal and can be published after a few corrections.
1. While all other spectra look pretty good, 1H NMR spectra of compounds 19b, 20b, 24a, 24b, 26a contain some unpurities.
The 1H NMR spectra of the samples for the compounds 19b, 20b, 24a, 24b, 26a, containing no impurities were recorded and are now included in the Supplementary Material.
2. On the line 531 it was mentioned that TMS is used as internal standard, but on the next line written that chemical shifts corrected on the basis of residual solvent peaks. I think that one of these thesises should be removed as contradictory. Generally, if TMS added it uses as internal standard to correct chemical shifts.
As TMS is an integral part of the deuterated dimethyl sulfoxide (DMSO), TMS was used as an internal standard. This is now clarified and the misleading sentence in the Materials and Methods section is omitted (p. 19, lines 553-554).
3. On the scheme 4 in the conditions (i) should form not the benzo[d]thiazole but its dihydro derivative: 2,3-dihydrobenzo[d]thiazole. How do you explain this? Is there is an oxidation with oxygen? Do you observe the intermediate dihydro product? Or some oxidant was lost in procedure? On the scheme 1 for this purpose Na2S2O5 was used (conditions iii).
The last step of cyclization of amidino-substituted 2-aminobenzenethiolate and corresponding benzaldehyde in glacial acetic acid and subsequent acid-base reaction was performed in accord with procedure described in the literature (Bioorg. Chem. 2020, 95, 103537) affording novel 6-imidazolyl 38a–38c, 39a, 39c, 40a–40c, 41a–41c and 6-pirimidinyl 42a, 42b, 43c, 44a–44c, 45a, 45c benzothiazoles, as shown in Scheme 4.
The indicated reference describing in details application of different methods in cyclization for the synthesis of benzothiazoles is now added in the manuscript. The formation of dihydro derivatives, 2,3-dihydrobenzo[d]thiazoles, was not observed. Therefore, we may assume that spontaneous oxidation with oxygen from the air in the cyclization of amidino-2-aminothiophenole with benzaldehyde occurred.